# Structural basis of tRNA recognition by the m³C RNA methyltransferase METTL6 in complex with SerRS seryl-tRNA synthetase

Philipp Throll[1,4], Luciano G. Dolce [1,4], Palma Rico-Lastres[2], Katharina Arnold [2], Laura Tengo [1], Shibom Basu [1], Stefanie Kaiser[3], Robert Schneider[2] & Eva Kowalinski [1] ✉

Methylation of cytosine 32 in the anticodon loop of tRNAs to 3-methylcytosine (m³C) is crucial for cellular translation fidelity. Misregulation of the RNA methyltransferases setting this modification can cause aggressive cancers and metabolic disturbances. Here, we report the cryo-electron microscopy structure of the human m³C tRNA methyltransferase METTL6 in complex with seryl-tRNA synthetase (SerRS) and their common substrate tRNA^Ser. Through the complex structure, we identify the tRNA-binding domain of METTL6. We show that SerRS acts as the tRNA^Ser substrate selection factor for METTL6. We demonstrate that SerRS augments the methylation activity of METTL6 and that direct contacts between METTL6 and SerRS are necessary for efficient tRNA^Ser methylation. Finally, on the basis of the structure of METTL6 in complex with SerRS and tRNA^Ser, we postulate a universal tRNA-binding mode for m³C RNA methyltransferases, including METTL2 and METTL8, suggesting that these mammalian paralogs use similar ways to engage their respective tRNA substrates and cofactors.

Transfer RNAs play a central role in connecting messenger RNA codons to their corresponding amino acids, and high-fidelity tRNA biogenesis is essential for accurate translation. Numerous nucleotide modifications ensure their correct folding and stability[1], protect them from nuclease cleavage[2] and ensure their correct subcellular localization[3]. Moreover, nucleobase modifications in the tRNA anticodon loop expand the coding capacity of tRNAs, assure accurate decoding and enhance translational fidelity and efficiency[4–7]. 3-Methylcytosine (m³C), which is present on base $C_{32}$ in the anticodon loop of the majority of serine, threonine and a subset of arginine isoacceptor tRNAs (tRNA^Ser, tRNA^Thr and tRNA^Arg), is one of these nucleobase modifications[8–10]. In the absence of $m^3C_{32}$, translation efficiency and fidelity are impaired[11–14].

In *Saccharomyces cerevisiae*, a single *S*-adenosyl-methionine (SAM)-dependent methyltransferase TRM140 (formerly ABP140) is responsible for $m^3C_{32}$ on tRNA^Thr and tRNA^Ser (refs. 12,15,16), while in the fission yeast *Schizosaccharomyces pombe* two distinct enzymes, TRM140 and TRM141, target either threonine or serine isoacceptors, respectively[17]. In mammals, three nonessential methyltransferases catalyze m³C on specific sets of tRNAs: METTL2A in cytosolic tRNA^Thr and tRNA^Arg, METTL6 in cytosolic tRNA^Ser and METTL8 in mitochondrial (mt)-tRNA^Thr_UGU and mt-tRNA^Ser_UGA [11,18–22]. These m³C RNA methyltransferases rely on cofactors to recognize and methylate their substrates, or require pre-existing tRNA modifications for cofactor-independent substrate recognition. The molecular mechanisms of their substrate selection remain unclear.

In this study, we focused on METTL6, which is expressed in all tissues of the human body (https://www.proteinatlas.org/ENSG00000206562-METTL6). The enzyme has been found

[1]European Molecular Biology Laboratory, Grenoble, France. [2]Institute of Functional Epigenetics, Helmholtz Zentrum Munich, Neuherberg, Germany. [3]Institute of Pharmaceutical Chemistry, Goethe University Frankfurt, Frankfurt, Germany. [4]These authors contributed equally: Philipp Throll, Luciano G. Dolce. ✉e-mail: kowalinski@embl.fr

overexpressed in patients with highly proliferative luminal breast tumors and it is linked to poor prognosis in hepatocarcinoma[11,23–28]. METTL6-deficient mice display lower metabolic rates and reduced energy turnover, and develop diverticular diseases[11,29,30]. The methylation activity of METTL6 is specific for $C_{32}$ of tRNA[Ser]. While there are reports of in vitro activity of isolated METTL6 (refs. [11,31]), other studies suggested that METTL6 activity depends on seryl-tRNA synthetase (SerRS)[18,19]. Nevertheless, clear evidence for a direct interaction between both proteins is lacking, since their association was sensitive to RNase treatment[18,19]. Recent crystal structures of METTL6 show how the enzyme interacts with the co-substrate SAM and the reaction co-product S-adenosyl-homocysteine (SAH), but allowed only speculations about how METTL6 binds tRNA or interacts with SerRS[31,32].

Here, we report the cryo-electron microscopy (cryo-EM) structure of METTL6 bound to tRNA[Ser] and SerRS at 2.4 resolution. Our data show how the long variable arm of the tRNA[Ser] is recognized by SerRS and anchors the tRNA in the interface between SerRS and METTL6. In addition, we reveal that METTL6 coordinates the tRNA anticodon arm through a domain that is common to all m³C RNA methyltransferases and remodels the anticodon stem loop into a configuration that flips out the target nucleotide $C_{32}$, rendering the base accessible for catalysis. This m³C tRNA methylation complex reveals a new moonlighting function of an aminoacyl-tRNA synthetase specifically targeting its bona fide set of substrates for chemical base modification.

## SerRS boosts the catalytic activity of METTL6

We aimed to resolve the ambiguity about METTL6 dependence on SerRS for m³C methylation. For this, we expressed full-length human METTL6 and SerRS individually in *Escherichia coli* and purified both proteins to homogeneity (Fig. 1a and Extended Data Fig. 1a–c). In our experiments, METTL6 was active on in-vitro-transcribed tRNA[Ser]$_{UGA}$ but not on tRNA[Thr]$_{CGU}$, which was used as a control. After adding SerRS to the reaction, the methyltransferase activity of METTL6 increased by approximately 1,000-fold. ATP and serine, the substrates of the aminoacylation reaction, were not necessary for the stimulation of METTL6 activity by SerRS (Fig. 1b and Extended Data Fig. 1d). We thus conclude that SerRS acts as an activation factor of METTL6 that strongly increases methylation activity.

## METTL6 and SerRS form a tRNA-dependent complex

We aimed to assemble the METTL6–SerRS–tRNA complex to define how METTL6 recognizes tRNA[Ser] and understand how METTL6 is activated by SerRS. Using size-exclusion chromatography (SEC), we did not observe complex formation between purified METTL6 and SerRS. But the addition of tRNA[Ser]$_{UGA}$ led to the reconstitution of a complex containing all components (Extended Data Fig. 1e). Despite attempts to crystallize the complex, we could only achieve two crystal forms of METTL6 bound to SAH (Extended Data Fig. 1f and Table 1) and the complex disintegrated during cryo-EM grid preparation. Therefore, for the purpose of structure determination, we stabilized the assembly by fusing METTL6 to the flexible C terminus of SerRS and coexpressed both proteins as a single polypeptide in insect cells. Under low-stringency purification conditions, the SerRS–METTL6 fusion construct was co-purified with cellular tRNA and subjected to cryo-EM structure solution (Extended Data Fig. 2a).

## Structural analysis of the METTL6–SerRS–tRNA[Ser] complex

Processing of the cryo-EM micrographs and particle images of the METTL6–SerRS–tRNA[Ser] complex revealed three-dimensional (3D) classes for 1:2:1, 1:2:2 and 2:2:2 stoichiometries with essentially identical intermolecular contacts in the METTL6–SerRS–tRNA interface (Extended Data Figs. 2b and 3a–c). This heterogeneity is partially due to the fact that SerRS forms an obligatory dimer that can bind either

one or two tRNAs[33–37]. Surprisingly, despite the fusion of METTL6 to SerRS, not all of the SerRS molecules in our reconstructions displayed an associated density for METTL6. In the absence of protease treatment or cleavage sites, this suggests that the 60 amino acid-long flexible linker connecting the proteins allowed free dissociation of METTL6 and that the fusion strategy tolerated an on–off equilibrium of the assembly. For model building and interpretation, we focused on the highest resolution class at 2.4 Å, resolving a 1:2:2 METTL6–SerRS–tRNA complex (Table 2).

In brief, at the core of the 1:2:2 METTL6–SerRS–tRNA complex, two molecules of tRNA[Ser] bind symmetrically across the SerRS dimer without contact with each other. Each tRNA[Ser] interacts with both SerRS protomers (Fig. 1c,d). METTL6 is located at the extremity of the complex, binding its proximal tRNA[Ser] anticodon stem and the variable arm that protrudes from the SerRS–tRNA[Ser] complex. The tRNA[Ser] long variable arm is coordinated in the interface between METTL6 and the SerRS N-terminal stalk. In the same interface, two protein loops of SerRS interact directly with METTL6.

## Insights into tRNA[Ser] recognition by human SerRS

SerRS selects serine isoacceptors by binding their characteristic long variable arm with a helix of the N-terminal domain (NTD) stalk in the cleft between the T-arm and the long variable arm (Fig. 1c,d)[33–39]. In contrast to previous human SerRS–tRNA[Sec] complex structures[37], our METTL6–SerRS–tRNA[Ser] structure revealed a state in which the tRNA is fully engaged with the stalk of SerRS, resembling rather the tRNA-bound crystal structure of the *Thermus thermophilus* enzyme (Supplementary Fig. 1a)[33]. Therefore, our structure reveals unprecedented details of how human SerRS binds the variable arm of tRNA[Ser] via nucleotide $U_{47B}$ and discloses how the tRNA geometry-defining 19-56 base pair is recognized (Supplementary Fig. 1b,c). The cryo-EM map does not resolve the full 3′ end of the acceptor arm, indicating that it is not engaged in the aminoacylation active site (Supplementary Fig. 1d). Therefore, in addition to the insights gained into the METTL6–SerRS–tRNA[Ser] interaction, the complex structure reveals new details of the binding of eukaryotic SerRS to tRNA[Ser].

## The m³C-specific tRNA-binding domain of METTL6 is conserved to the m³C RNA methyltransferase family

METTL6 shares the Rossmann-like fold of class I methyltransferases[40,41]. The family of m³C-specific RNA methyltransferases (Interpro IPR026113) have divergent N-terminal sequences and a conserved insertion within the Rossman fold, both of undetermined function (Extended Data Fig. 4). Our structure of the METTL6–SerRS–tRNA[Ser] complex pinpoints that three m³C methyltransferase-specific elements form a tripartite domain consisting of the N-terminal region, the internal insertion and the extended hairpin of the central β-sheet. All elements of this 'm³C methyltransferase-specific RNA-binding domain' (m³C-RBD) contribute to RNA binding. (Fig. 2a,b). The domain forms a highly positively charged groove that accommodates the anticodon stem through base-unspecific electrostatic contacts with the phosphodiester backbone; only the modification base $C_{32}$ is recognized through base-specific contacts (Fig. 2c).

In the N-terminal element of the m³C-RBD, two helices form the interface with SerRS in the complex. While the sequence of the first helix (αN1) is not conserved across the m³C methyltransferase family members, the second helix (αN2) carries the conserved WDLFYK motif, which we identify as playing a central and multivalent role in coordinating the interactions with SerRS, SAH and the variable arm of the tRNA. These helices are followed by an unstructured linker region carrying the strongly conserved FFKDRHW motif, which we term the 'substrate-binding loop', and which conveys a negative charge to the binding cleft. The substrate-binding loop is central in coordinating the target base $C_{32}$ and contributes to the coordination of SAH

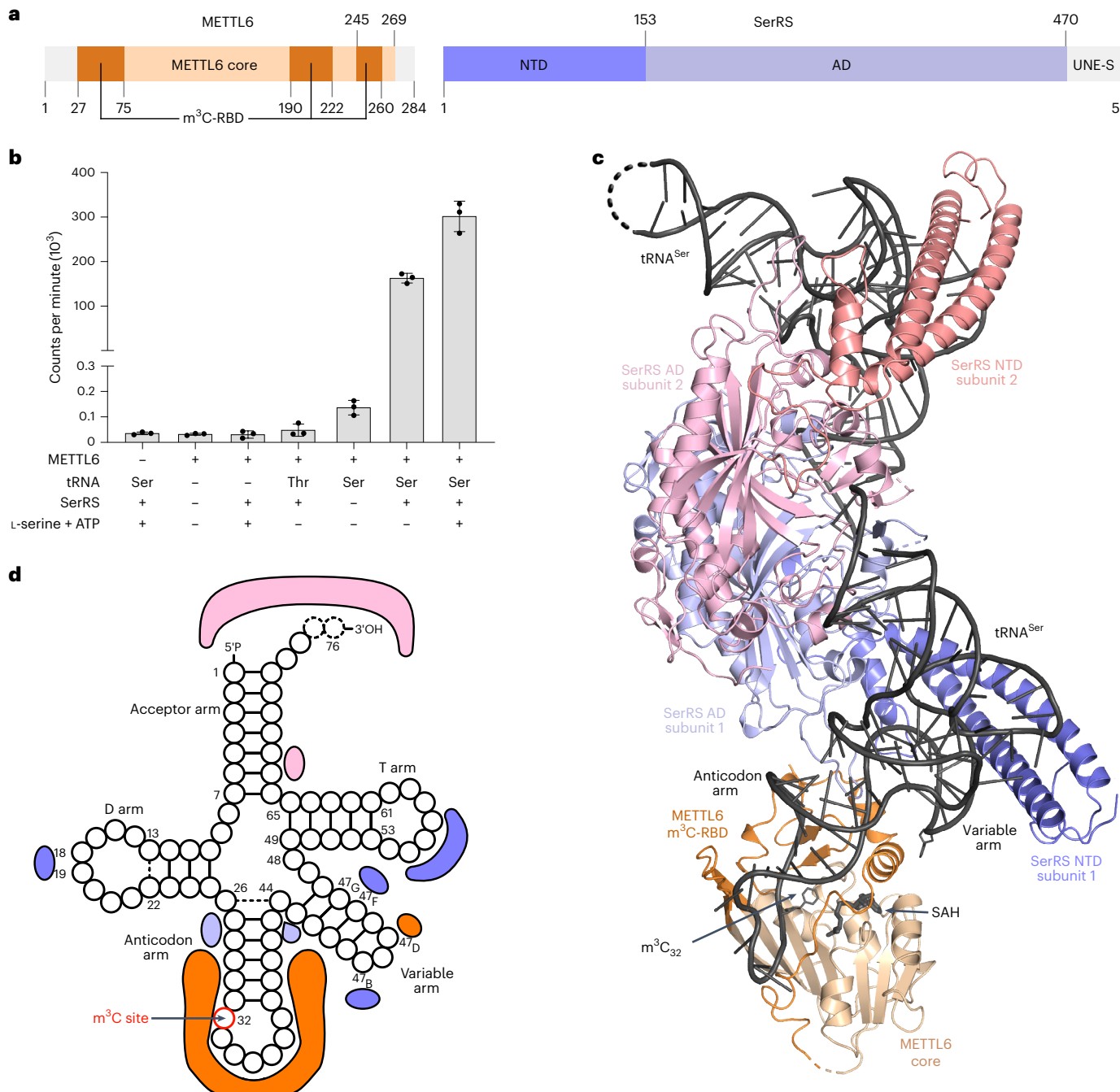

**Fig. 1 | The structure of the SerRS–tRNA–METTL6 complex. a**, Domain organization of human METTL6 and SerRS. The methyltransferase core of METTL6 is colored in light orange and the RNA-binding domain (m³C-RBD) in dark orange. The NTD of SerRS is colored in dark purple and the AD in light purple. Portions that are unresolved in the structure are colored in gray (N and C terminus of METTL6, UNE-S of SerRS). **b**, In vitro methylation activity of METTL6 in the presence and absence of SerRS. Control experiments in

Extended Data Fig. 1d. Data are represented as the blank-subtracted mean and s.d. of independent replicates (*n* = 3). **c**, The structural model of the 1:2:2 METTL6–SerRS–tRNA complex based on the cryo-EM reconstruction (protein domains are colored as in **a**); in the second SerRS protomer the NTD is colored in salmon and the AD in light pink; tRNA and SAH are colored in black. **d**, Schematic depicting the protein contacts of METTL6 and SerRS to the commonly coordinated tRNA molecule. Colors as in **c**.

(Fig. 2d). The second element of the m³C-RBD is an insertion embedded between β5 and αE of the methyltransferase core, consisting of a short helical turn and a β-hairpin, which both bind to the anticodon stem (Fig. 2b,d,e). The third element of the m³C-RBD consists of a hairpin structure formed by the unusually elongated β-strands, β6 and β7, which we term β-extension. This hairpin region carries mostly positively charged residues, and the cryo-EM map indicates some degree of flexibility of this motif, possibly to enable the accommodation of diverse sequences in the variable anticodons of serine tRNAs (Fig. 2e).

Surprisingly, the topological arrangement of the m³C-RBD of METTL6 did not resemble any other substrate-binding domains in characterized RNA or DNA methyltransferases. Among similarity search results, we identified the methyltransferases METTL11A and METTL11B, which methylate the N-terminal α-amino group of proteins, as relatives with a similar topology (Extended Data Fig. 5a,b)[41–46]. Overall, our METTL6–SerRS–tRNA^Ser complex structure identifies the m³C methyltransferase-specific RNA-binding domain, which differs from so-far characterized DNA or RNA methyltransferases.

**Table 1 | X-ray crystallographic data collection and refinement statistics**

| | METTL6 (full length) (PDB 8OWX) | METTL6 (40–269) (PDB 8OWY) |
|---|---|---|
| **Data collection** | | |
| Space group | $P4_3$ | $C2$ |
| Cell dimensions | | |
| $a, b, c$ (Å) | 82.856, 82.856, 130.39 | 81.971, 138.129, 39.66 |
| α, β, γ (°) | 90, 90, 90 | 90, 115.257, 90 |
| Resolution (Å) | 41.43–2.601 (2.694–2.601) | 39.11–3.2 (3.315–3.2) |
| $R_{sym}$ or $R_{merge}$ | 0.2006 (4.8) | 0.2055 (2.196) |
| $I / \sigma(I)$ | 9.43 (0.45) | 6.97 (0.80) |
| Completeness (%) | 98.36 (83.61) | 85.31 (48.00) |
| Redundancy | 13.6 (13.3) | 6.5 (6.2) |
| **Refinement** | | |
| Resolution (Å) | 41.43–2.601 (2.694–2.601) | 39.11–3.2 (3.315–3.2) |
| No. reflections | 26,552 (2,219) | 5,634 (312) |
| $R_{work} / R_{free}$ | 0.2170 (0.3804)/ 0.2503 (0.3708) | 0.2603 (0.3426)/ 0.2990 (0.4826) |
| No. atoms | 4,005 | 2,673 |
| Protein | 3,936 | 2,623 |
| Ligand/ion | 69 | 50 |
| Water | 0 | 0 |
| $B$ factors | | |
| Protein | 86.58 | 51.66 |
| Ligand/ion | 77.30 | 46.26 |
| Water | N/A | N/A |
| R.m.s. deviations | | |
| Bond lengths (Å) | 0.012 | 0.013 |
| Bond angles (°) | 1.85 | 2.03 |

Values in parentheses are for highest resolution shell. N/A, not available.

## tRNA$^{Ser}$ binding induces structural rearrangement in the m$^3$C-RBD

To determine whether tRNA binding to METTL6 induces conformational changes in the enzyme, we compared our structural model to the crystal structures of METTL6 bound to either SAM or SAH, but without an RNA ligand[31,32]. The methyltransferase core of all models aligns well (Supplementary Table 1). Deviations occur exclusively in the m$^3$C-RBD, in particular in the divergent N-terminal region, the substrate-binding loop and the β-extension. All RNA-free structures display disordered substrate-binding loops, which seem to adopt their shape only in contact with the tRNA, and a conserved helix following the substrate-binding loop is rearranged on interaction with the ligand. Our data suggest some degree of flexibility in the region of the β-extension hairpin, a region that is completely unresolved in the crystal structures, suggesting its plasticity for tRNA$^{Ser}$ accommodation (Fig. 2e). Thus, while the core of METTL6 is rigid, parts of the m$^3$C-RBD seem to be more ductile, to accommodate the tRNA.

## The METTL6-bound tRNA anticodon arm is remodeled for exposure of C$_{32}$

Analysis of the cryo-EM map in the tRNA regions suggested that our sample contains a mixture of different serine isoacceptors. Accordingly, we built a tRNA$^{Ser}$ consensus model based on the cryo-EM map density, tRNA$^{Ser}$ sequence conservation and the conservation of base modifications (Extended Data Fig. 6a–d)[8]. Comparing the METTL6-bound tRNA$^{Ser}$ to the METTL6-unbound tRNA$^{Ser}$ copy in our structure, and also to free tRNA, we observe extensive remodeling of the anticodon arm and its global bending toward METTL6 (Fig. 3a and Extended Data Fig. 5c). In free tRNA, C$_{32}$ is generally engaged in a stable noncanonical base pair with A$_{38}$ (ref. 47). Strikingly, in the complex structure, this C$_{32}$-A$_{38}$ base pair is disrupted and the bases are flipped to the outside of the anticodon loop. Such a flip-out mechanism as a prerequisite for base modification is a common feature in RNA-modifying enzymes, such as ADAT and ADAR deaminases[48–50]. Importantly, the extensive anticodon loop rearrangement brings C$_{32}$ into the proximity of the methyl-donor molecule bound in the METTL6 methyltransferase core and makes the base accessible for methyl transfer.

## Coordination of the A$_{37}$ modification

The cryo-EM map revealed the typical modifications of eukaryotic tRNA$^{Ser}$ (Extended Data Fig. 6)[8]. Interestingly, the modification of adenosine A$_{37}$ in the anticodon loop forms contacts to the m$^3$C-RBD. Mass spectrometry identified predominantly i$^6$A in the copurified tRNA$^{Ser}$ (Extended Data Fig. 6b,c). In the complex structure, the i$^6$A$_{37}$ nucleobase stacks with the last base pair of the anticodon stem (A$_{31}$-U$_{39}$); the modified isopentenyl moiety is coordinated by histidine H195, threonine T248 and two conserved arginine residues (R199 and R259), all located in the m$^3$C-RBD (Fig. 3b). The premodification of adenosine A$_{37}$ to N6-isopentenyladenosine (i$^6$A$_{37}$) or N6-threonylcarbamoyladenosine (t$^6$A$_{37}$) has been described as a prerequisite for m$^3$C$_{32}$ modification in threonine tRNAs by METTL2A and METTL8 and the related yeast enzymes[16,17,19–21]. Although the A$_{37}$ modification was not a requirement for METTL6 activity[11,19], the coordination of this modification in our complex structure suggests that i$^6$A$_{37}$ might assist in stabilizing the remodeled tRNA$^{Ser}$ configuration. The two isopentenyl-binding arginine residues (R199 and R259) are conserved throughout the m$^3$C methyltransferase family, suggesting that the METTL6 paralogs would coordinate i$^6$A$_{37}$ or t$^6$A$_{37}$ in a similar fashion, and possibly exhibit hydrogen bonding toward t$^6$A$_{37}$ (Extended Data Fig. 7).

The cryo-EM map density suggested the presence of SAH in our reconstruction. As in RNA-free crystal structures of METTL6 (refs. 31,32), the methyl donor is embedded in a deep pocket adjacent to the RNA-binding cleft in METTL6 (Fig. 4a). Motif I of methyltransferases (EaGCGvGN in METTL6) and the main chain carbonyl of isoleucine I157 interact with the amino acid portion of SAH; arginine R60 forms a salt bridge with the carboxyl group of the cofactor. Aspartate D110 in the acidic loop of motif II, which is highly conserved in SAM-dependent methyltransferases, maintains hydrogen bonds to the sugar hydroxyl groups of SAH, and also tryptophan W45 and valine V159 contribute to the sugar coordination. Phenylalanine F111 of the acidic loop and other mostly hydrophobic residues (A162, V163, F111, L137, T138) form the pocket for the adenosyl moiety of SAM (Fig. 4b and Extended Data Fig. 8).

Mass spectrometry of the copurified tRNA confirmed the presence of m$^3$C-modified tRNA$^{Ser}$ in our sample (Extended Data Fig. 6b). In the structural model, the modified N3 atom of C$_{32}$ lies at a 4 Å distance from the sulfur atom of SAH. These findings indicate that the structure captured a postcatalytic state with both products present (Fig. 3a). The m$^3$C$_{32}$ nucleotide is tightly coordinated in the interface of the methyltransferase core and the m$^3$C-RBD (Fig. 4c and Extended Data Fig. 8). The bottom of the C$_{32}$-binding pocket is formed by aromatic residues of the methyltransferase core, that is, tyrosine Y190 and phenylalanines F261 and F158. Interestingly, F158 lies in the position where some other SAM-dependent RNA and DNA methyltransferases would carry the catalytic (D/N)PP(F/W/Y) motif[40]. Phenylalanines F158 (methyltransferase core) and F57 (m$^3$C-RBD) stack the C$_{32}$ base via π-interactions. Two opposed arginine residues interact from each side with functional groups of the base R60 via hydrogen bonding with the carbonyl and R213 with the amino group, suggesting a deprotonated state. Several

**Table 2 | Cryo-EM data collection, refinement and validation statistics**

| | METTL6–SerRS–tRNA, 1: 2: 2 (EMD-17528), (PDB 8P7B) | METTL6–SerRS–tRNA, 2: 2: 2 (EMD-17529), (PDB 8P7C) | METTL6–SerRS–tRNA, 1: 2: 1 (EMD-17530), (PDB 8P7D) | SerRS–tRNA, 2: 1 (EMD-17531) | Free tRNA (EMD-17532) |
|---|---|---|---|---|---|
| **Data collection and processing** | | | | | |
| Magnification | 130,000 | 130,000 | 130,000 | 130,000 | 130,000 |
| Voltage (kV) | 300 | 300 | 300 | 300 | 300 |
| Electron exposure (e⁻/Å²) | 63.27 | 63.27 | 63.27 | 63.27 | 63.27 |
| Defocus range (µm) | –0.8 to –1.8 | –0.8 to –1.8 | –0.8 to –1.8 | –0.8 to –1.8 | –0.8 to –1.8 |
| Pixel size (Å) | 0.645 | 0.645 | 0.645 | 0.645 | 0.645 |
| Symmetry imposed | $C1$ | $C2$ | $C1$ | $C1$ | $C1$ |
| Initial particle images (no.) | 1,625,946 | 1,625,946 | 1,625,946 | 1,625,946 | 1,625,946 |
| Final particle images (no.) | 460,853 | 105,928 | 35,788 | 49,372 | 112,675 |
| Map resolution (Å) | 2.4 | 3.7 | 4.2 | 3.8 | 5.6 |
| FSC threshold | 0.143 | 0.143 | 0.143 | 0.143 | 0.143 |
| Map resolution range (Å) | 2.2–3.4 | 3.0–7.0 | 4.0–8.0 | 3.3–5.7 | 5.0–7.0 |
| **Refinement** | | | | | |
| Initial model used (PDB code) | 4RQE, 8OWX | 8P7B | 8P7B | | |
| Model resolution (Å) | 2.4 | 3.7 | 4.2 | | |
| FSC threshold | 0.143 | 0.143 | 0.143 | | |
| Model resolution range (Å) | 2.2–3.4 | 3.0–7.0 | 4.0–8.0 | | |
| Map sharpening $B$ factor (Å²) | –65.1 | –121.2 | –101.8 | | |
| Model composition | | | | | |
| Nonhydrogen atoms | 12,363 | 14,275 | 9,686 | | |
| Protein residues | 1,143 | 1,383 | 1,008 | | |
| Ligands | 1 | 2 | 1 | | |
| $B$ factors (Å²) | | | | | |
| Protein | 77.74 | 215.93 | 310.45 | | |
| Ligand | 103.19 | 252.95 | 379.51 | | |
| R.m.s. deviations | | | | | |
| Bond lengths (Å) | 0.024 | 0.024 | 0.021 | | |
| Bond angles (°) | 1.767 | 1.755 | 1.557 | | |
| Validation | | | | | |
| MolProbity score | 1.58 | 1.95 | 1.59 | | |
| Clashscore | 7.19 | 15.95 | 6.68 | | |
| Poor rotamers (%) | 0.44 | 1.02 | 0.50 | | |
| Ramachandran plot | | | | | |
| Favored (%) | 96.89 | 96.40 | 96.58 | | |
| Allowed (%) | 3.11 | 3.60 | 3.42 | | |
| Disallowed (%) | 0.00 | 0.00 | 0.00 | | |

other residues lie at distances that would allow van der Waals interactions with $C_{32}$, such as tryptophan W45 and tyrosine Y190. Adjacent to $C_{32}$, the phosphodiester backbone of the RNA is stabilized through interactions with tyrosine Y190, arginine R259, phenylalanine F261 and tryptophan W62.

To probe the METTL6 active site experimentally, we tested single-point mutants for enzymatic activity, for methyl-donor and tRNA$^{Ser}_{UGA}$ binding and for complex formation (Fig. 4d and Extended Data Fig. 9a–d). SAM-dependent methyltransferases use diverse reaction mechanisms[51]. The absence of cysteines in proximity to $C_{32}$ excludes the possibility of covalent enzyme–nucleotide intermediate formation with the base, as has been described for the uracil-5-methyltransferase TrmA[52]. Glutamates or aspartates that could be potentially catalytically

relevant are not found in the close environment of $C_{32}$. Therefore, we tested mutations of two conserved tyrosines (Y49 and Y190; Fig. 4b,c) in proximity to $C_{32}$ as potential candidates that could play a role in enzyme activity. For methyl-donor binding, we used thermal stability assays in the presence of SAH, since this ligand resulted in the largest stability increase (Extended Data Fig. 9b). To probe the tRNA binding of METTL6 alone, we conducted fluorescence polarization (FP) experiments in the absence of SerRS (Extended Data Fig. 9c). The mutation of aspartate D110 to alanine (D110A), a highly conserved and indispensable SAM-coordination residue in this class of methyltransferases[41], resulted in the loss of SAH binding; it had impaired tRNA$^{Ser}_{UGA}$ affinity and was not active. The Y49F mutation, a tyrosine with a putative role in SAM binding, could still coordinate the methyl donor, but showed

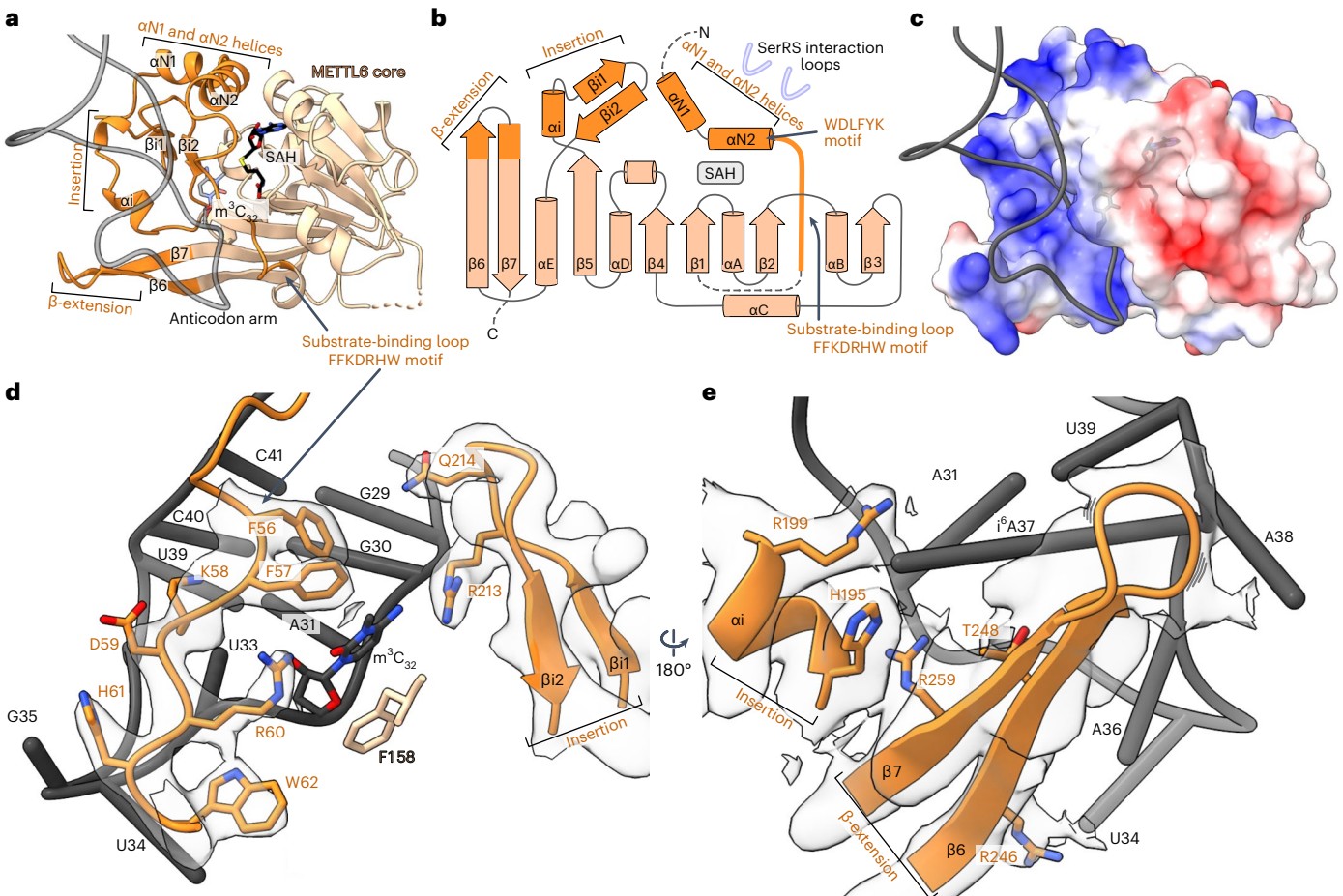

**Fig. 2 | The m³C-specific RNA-binding domain (m³C-RBD) of METTL6.**
**a**, Cartoon model of METTL6 from the complex structure with the methyltransferase core in light orange and the m³C-RBD in dark orange. The phosphodiester backbone of the anticodon arm of the tRNA is in gray; $C_{32}$ and SAH are represented as stick models. **b**, Rossmann fold scheme of METTL6. The three segments of m³C, the SerRS interacting region and the SAH-binding site are indicated. **c**, Surface charge distribution of METTL6 as calculated by ChimeraX[66] with same view as in **a**. Positive charges are indicated in blue, negative charges in red. **d**, The substrate-binding loop and β-turn part of the insertion contribute to the binding of the anticodon loop of the tRNA. Residues involved in $C_{32}$ binding and residues of the FFKDRHW motif are represented with side chains as sticks. **e**, The β-elongation and α-helical part of the insertion embrace the anticodon loop from the opposite side. The Coulomb density is shown at a level of 0.3 in a range of 3 Å around the m³C-RBD. Residues involved in $U_{34}$ or $i^6A_{37}$ binding are represented with side chains as sticks.

weakened tRNA$^{Ser}_{UGA}$ binding and attenuated activity. The mutation of tyrosine Y190 to phenylalanine (Y190F), in the proximity of m³C$_{32}$, could still bind tRNA$^{Ser}_{UGA}$, but its catalytic activity was attenuated. All mutants were impaired in complex formation on SEC (Extended Data Fig. 9d). As a side note, mutants of aspartate D189 to alanine or asparagine (D189A or D189N) were not soluble, suggesting a structural scaffolding role of this residue. We conclude that single-point mutations in the active site impair the complex formation of METTL6 with tRNA$^{Ser}$, the methyl donor and SerRS, resulting in attenuated methylation activity.

## Contacts between SerRS and METTL6 are indispensable for m³C$_{32}$ modification of tRNA$^{Ser}$

It was previously shown that METTL6 and also *Saccharomyces cerevisiae* TRM140 activities rely on the presence of the long variable arm of tRNA$^{Ser}$ (refs. 19,16), but it was unclear how METTL6 recognizes this arm. The METTL6–SerRS–tRNA$^{Ser}$ complex structure reveals that the variable arm of tRNA$^{Ser}$ is embedded in the interface between METTL6 and SerRS (Fig. 5a). METTL6 coordinates the strictly conserved tRNA$^{Ser}$ variable arm residue U$_{47D}$ through a hydrogen-bonding network involving conserved residues of the m³C-RBD, such as aspartate D46 and lysines K43 and K50; arginine R114, in the core of METTL6, also contacts the variable arm (Fig. 5b and Extended Data Fig. 10a). Nevertheless, when

we assessed tRNA binding in fluorescence polarization experiments, METTL6 alone could not distinguish tRNA$^{Ser}_{UGA}$ from other tRNA species (Extended Data Fig. 10b).

We hypothesized rather that SerRS selects the substrate and METTL6 recognizes a SerRS–tRNA$^{Ser}$ complex. But remarkably only two loops of SerRS interact directly with METTL6 (Fig. 5c,d). In the first contact point, the main chain of an NTD loop of SerRS (lysine K12 and glycine G13) contacts the side chain of phenylalanine F32 in the m³C-RBD (Fig. 5c and Extended Data Figs. 3a and 10c). The second SerRS interface loop, which is part of the aminoacylation domain (AD), forms an intricate interaction network with several elements of the METTL6 m³C-RBD, and also contacts the upper part of the tRNA anticodon stem (Fig. 5d and Extended Data Fig. 10d). Here, polar interactions of two SerRS AD methionines, M416 and M417, engage with METTL6 m³C-RBD elements, particularly via arginine R51 and main chain interactions to the β-hairpin in the β-insertion (residues Q214–G216). Mutation of the METTL6 phenylalanine F32 to alanine (F32A) prevented complex formation in SEC and attenuated methylation activity (Fig. 5e and Extended Data Fig. 10e). The double mutant of the SerRS methionines to alanines (MM416/417AA) disrupted the complex and suppressed methylation activity (Fig. 5e and Extended Data Fig. 10e). It is important to note that the METTL6–SerRS interface residues phenylalanine F32 and arginine R51 are exclusive to m³C methyltransferases that

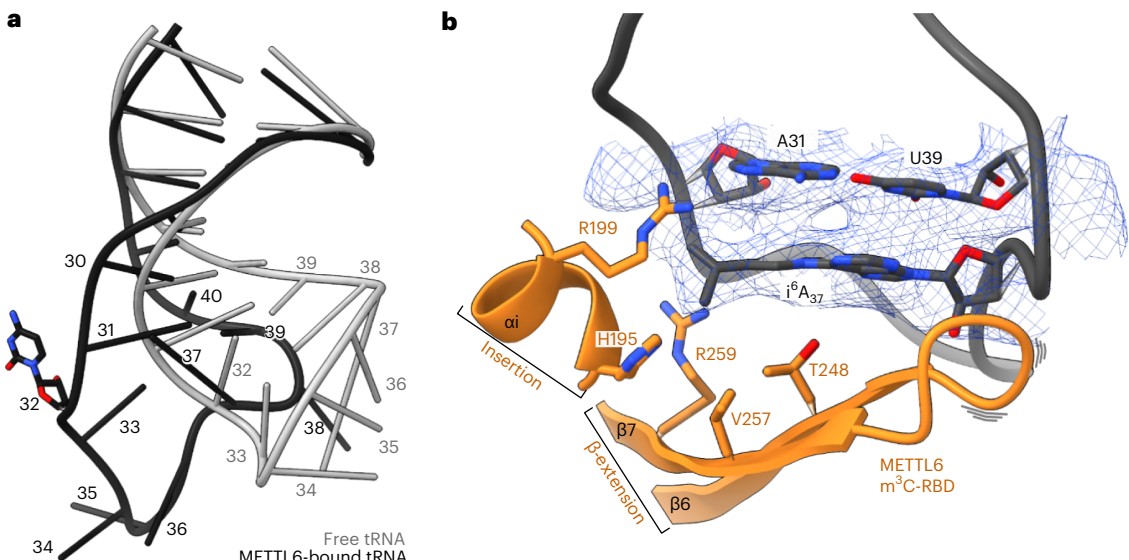

**Fig. 3 | Remodeling of the tRNA anticodon arm. a**, The tRNA anticodon arm is remodeled in the METTL6–SerRS–tRNA complex (black) in comparison to free tRNA (PDB 1EHZ, gray) (r.m.s.d. = 1.124 Å). **b**, The modification of adenosine A37 ($i^6A_{37}$) in the tRNA is coordinated through conserved residues in the $m^3$C-RBD. Residues involved in $i^6A_{37}$ coordination are represented with side chains as sticks. The Coulomb density is shown at a level of 0.3 in a range of 2 Å around the $m^3$C-RBD.

target cytosolic tRNA$^{Ser}$ and that the SerRS contact loops are conserved as well (Fig. 5f and Extended Data Fig. 10g,h). Thus, SerRS is the substrate-recognition factor for METTL6 and the conserved direct contacts between METTL6 and SerRS are indispensable for the coordinated binding of the variable arm of tRNA$^{Ser}$.

## Discussion

In this work, we present the cryo-EM structure of METTL6 in complex with SerRS bound to the co-products of the methyl transfer, the $m^3C_{32}$-modified tRNA$^{Ser}$ and SAH. We identify the tripartite tRNA-binding domain of METTL6 and show that SerRS is necessary as a tRNA selection factor that augments METTL6 activity.

Aminoacylation-independent functions of aminoacetylases have been identified, most commonly as gene-regulatory auto-feedback mechanisms[53–56]. SerRS plays a noncanonical role in vascular development that is independent of its aminoacylation activity[57–59]. Here we reveal, to our knowledge, a new moonlighting function of an aminoacetylase as a substrate selection factor[60]. SerRS has high specificity for only tRNA$^{Ser}$, and as such facilitates very selectively the chemical base modification of its own set of isoacceptors.

We observe that the association of METTL6 to the SerRS–tRNA complex is rather labile, exemplified through the dissociation of METTL6 from the complex in cryo-EM samples, complex dissociation in higher dilutions (for example, on SEC columns with larger volume) and the interference of single-point mutations with complex formation (Figs. 4d and 5e). Therefore, we speculate that the involvement of SerRS does not necessarily provide a higher affinity toward tRNA, but rather that the complex provides the necessary geometry to trigger the reaction.

In our samples, we observe different possible stoichiometries of the METTL6–SerRS–tRNA complex. In the complex with two copies of METTL6 present, these are spatially distant (~100 Å) from each other, with each METTL6 molecule binding only one tRNA, implying that that these METTL6 molecules would function independently of each other. According to quantitative mass spectrometry, the cellular abundance of SerRS is approximately 100 times higher than METTL6 (ref. 61). Therefore, we would suppose that the most abundant physiological active assembly comprises the SerRS dimer, one or two tRNAs and one copy of METTL6, but does not exclude the association of a second METTL6 molecule to the complex.

It has been demonstrated that METTL6 activity does not depend on the serylation activity of SerRS[19]. Although we did not collect evidence on the order of the catalytic events under physiologic conditions, we would anticipate that serylation precedes methylation with the following reasoning: METTL6 is active on in-vitro-transcribed nonserylated tRNA; on the basis of cellular protein ratio with a large excess of SerRS in respect to METTL6, the tRNA is likely to encounter SerRS before interaction with METTL6; additionally, we observe the coordination of another RNA modification, $i^6A_{37}$ or $t^6A_{37}$, suggesting that $m^3C$ methylation is a late event in the tRNA biogenesis. The increased activity of METTL6, when ATP and serine are added to our assay (Fig. 1b), may be attributed to the favorable geometric alignment of the $m^3C$ methylation complex, which appears to be ideal for serylated tRNA$^{Ser}$, the probable physiologic substrate.

We identified the RNA-binding domain of $m^3C$ methyltransferases, $m^3C$-RBD. Functional key residues in the $m^3C$-RBD of METTL6 are conserved in the other family members, METTL2 and METTL8, and yeast enzymes, such as the SAM-binding motifs, the substrate-binding loop, the $C_{32}$-coordinating residues and residues that mediate the interaction with the $i^6A_{37}$ or $t^6A_{37}$ modification, respectively (Extended Data Figs. 4 and 7b–d). These similarities suggest resembling RNA-binding modes for all paralogs in the $m^3C$ RNA methyltransferase family. Indeed, AlphaFold2 models for the baker's and fission yeast homologs of METTL6 and the other human $m^3C_{32}$ methyltransferases, METTL8 and METTL2A (https://alphafold.ebi.ac.uk/), are similar to the tRNA-bound structure of METTL6 (Supplementary Table 1). However, they are distinct in their divergent N-terminal sequences. Our data show that the variable N-terminal region of METTL6 has a dual role in mediating cofactor-specific interactions with SerRS and in providing substrate-specific interactions to the long variable arm of tRNA$^{Ser}$. Analogously, METTL2 and METTL8 might carry specificity signals in their N termini, for example, for recruitment of mitochondrial SerRS2 to METTL8 for mt-tRNA$^{Ser}_{UGA}$ selection, or for recruitment of DALRD3 to METTL2B for tRNA$^{Arg}$ modification. Notably, DALRD3 is predicted to carry an anticodon-binding domain similar to class Ia aminoacyl-tRNA synthetases. Our discovery of the tripartite configuration of the $m^3C$-RBD shows that the three elements of the METTL6 $m^3C$-RBD act collectively for SerRS binding and tRNA$^{Ser}$ recognition. This observation clarifies why previous attempts to swap N termini between METTL6 and METTL2A in chimeric enzymes did not change their substrate specificities[19].

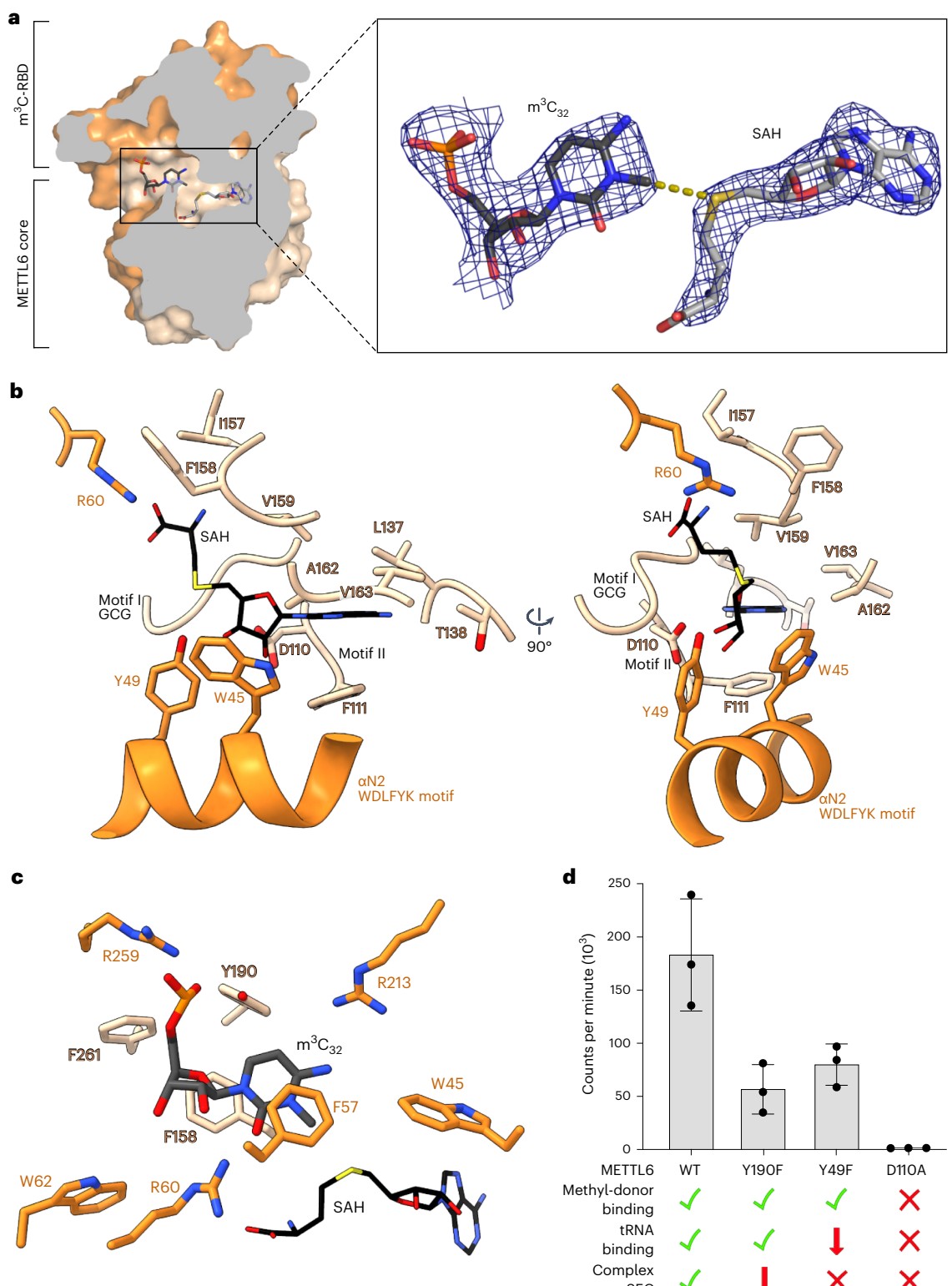

**Fig. 4 | The METTL6 active site. a**, Surface slice representation of METTL6 from the cryo-EM structure with the m3C32 and SAH in the active site, which lies in the interface between the methyltransferase core and m3C-RBD. The zoomed-in area shows the Coulomb density around m3C32 and SAH. **b**, SAH coordination in the interface of the methyltransferase core domain (pale orange) and m3C-RBD (dark orange) in two views. Residues within a 5 Å radius of SAH are represented with side chains as sticks. **c**, The m3C32-binding site adjacent to the SAH-binding site. Residues within a 5 Å radius of m3C32 are represented with side chains as sticks. **d**, Methyltransferase activity of single-point mutants in the active site with a qualitative indication of control experiments for complex formation assessed via SEC (Extended Data Fig. 9a), SAH binding by Thermofluor (Extended Data Fig. 9c) and tRNA binding by FP (Extended Data Fig. 9d). Methyltransferase control experiments in Extended Data Fig. 6b. Data are represented as the blank-subtracted mean and s.d. of independent replicates (*n* = 3). WT, wild type.

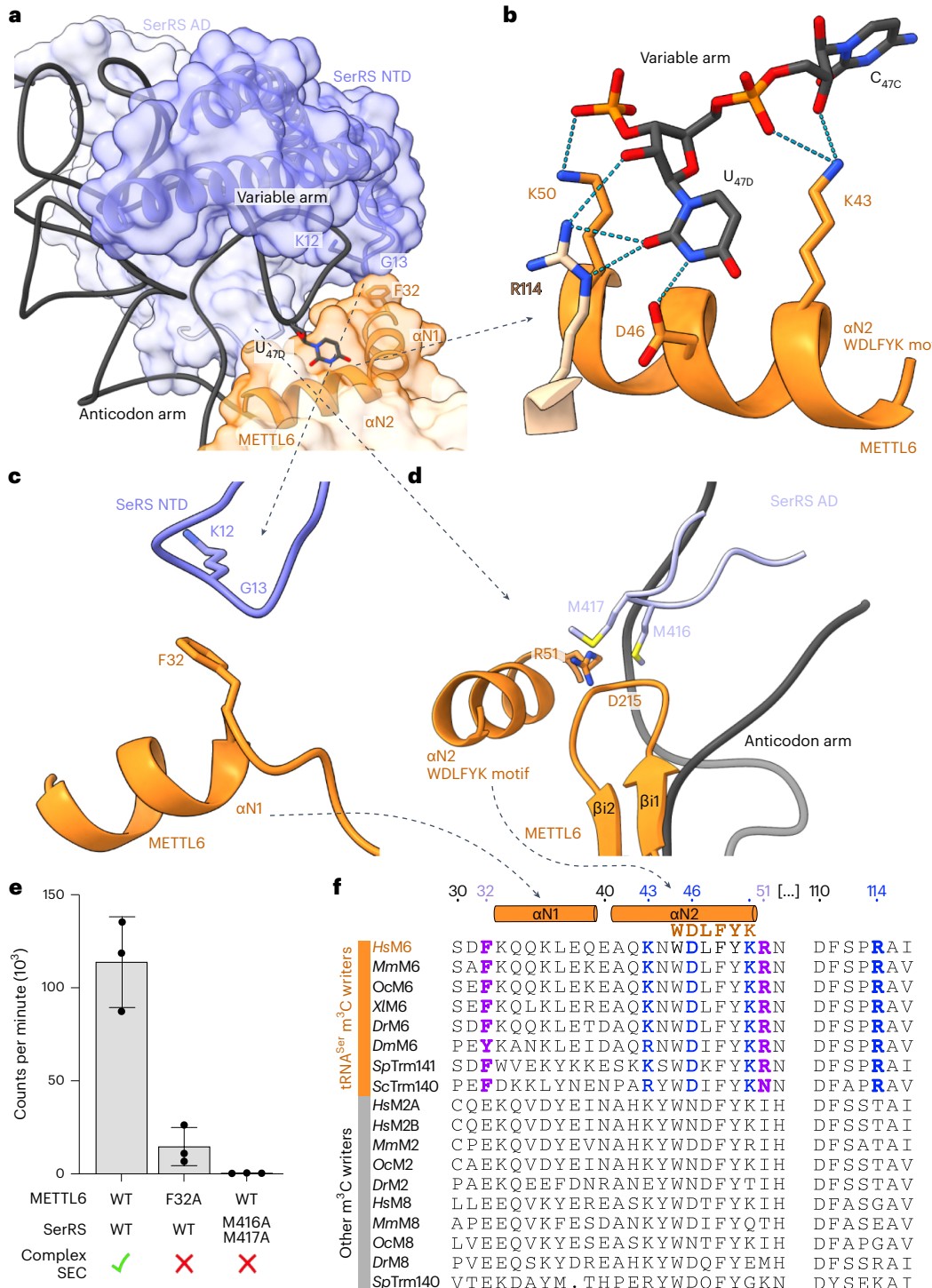

**Fig. 5 | The METTL6–SerRS–tRNA variable arm interface. a**, Cartoon and surface representation of the METTL6–SerRS–tRNA variable arm interface. SerRS NTD and αN1 and αN2 helices of METTL6 as cartoon representation with the F32 contact region to SerRS NTD loop indicated. U$_{47D}$ as stick representation. **b**, Variable arm contacts to METTL6, cartoon and stick representation, dotted lines represent hydrogen bonds. Residues involved in hydrogen with U$_{47D}$ are represented with side chains as sticks. **c**, Interaction of METTL6 F32 with the backbone of a SerRS NTD loop. **d**, Interaction of METTL6 m³C-RBD with SerRS AD interface loop. Cartoon and stick representation. Color scheme as in previous figures. Conserved residues within the METTL6–SerRS interface are represented with side chains as sticks. **e**, In vitro methyltransferase assay of interface mutants of METTL6 and SerRS with a qualitative indication of complex

formation assessed via SEC (Extended Data Fig. 10e). Methyltransferase assay control experiments in Extended Data Fig. 10f. Data are represented as the blank-subtracted mean and s.d. of independent replicates (*n* = 3). **f**, Multiple sequence alignment of the N-terminal helices and specific contact residues of METTL6 with other m³C methyltransferases. Residues involved in METTL6–SerRS contacts in purple, residues involved in METTL6–tRNA$^{Ser}$ variable arm contacts in blue. Bold letters signify residues exclusively conserved in m³C-RNA methyltransferases that target cytosolic tRNA$^{Ser}$. The species on the alignment are *Hs, Homo sapiens; Mm, Mus musculus; Oc, Oryctolagus cuniculus; Xl, Xenopus laevis; Dr, Danio rerio; Dm, Drosophila melanogaster; Sp, Schizosaccharomyces pombe* and *Sc, Saccharomyces cerevisiae.*

Although the genes for METTL2, METTL6 and METTL8 are nonessential, all three enzymes have been linked to disease. METTL6 acts as an oncogene in luminal breast tumors and hepatocarcinoma[11,23–28], and METTL2A also seems to play a role in breast cancer[62]. Similarly, the mitochondrial m³C methyltransferase METTL8 has been linked to severe pancreatic adenocarcinoma and colon cancer and is involved in neurogenesis[21,63–65]. Our data reveal how m³C methyltransferases interact with RNA. Hence we establish a foundation for future detailed investigations on the enzymes of this family, and also reveal promising prospects for potential therapeutic applications.

## Online content

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

## Methods

### Cloning, protein expression and purification in *E. coli*

Protein coding DNA sequences for METTL6 (UniProt Q8TCB7) with a 3C protease cleavable N-terminal 6×His-GST tag and SerRS (UniProt P49591) with a C-terminal 6×His-tag were cloned into a pEC vector. Point mutations of the proteins were produced by site-directed mutagenesis using single or double-mismatch primers.

All proteins were expressed in *E. coli* of the Rosetta II (DE3) strain and cultured in terrific broth medium. Expression was induced with 0.1 mM IPTG at 18 °C overnight. Cell pellets were frozen and kept at −80 °C before purification.

Wild type or mutant METTL6 was purified by resuspending and sonicating the cell pellet in lysis buffer containing 20 mM Tris-HCl, pH 7.5, 250 mM NaCl, 20 mM imidazole, 0.05% (v/v) 2-mercaptoethanol, 1 mM PMSF, 1 µg ml$^{-1}$ DNase I, 0.5 µg ml$^{-1}$ RNase and 50 µg ml$^{-1}$ lysozyme. The lysate was cleared by centrifugation with 20,000g followed by filtration with a 5 µM filter and loaded on a HisTrap HP column pre-equilibrated with lysis buffer (GE Healthcare). The column was washed first with lysis buffer, then with high-salt buffer (20 mM Tris-HCl, pH 7.5, 250 mM NaCl, 1 M KCl, 50 mM imidazole) and low-salt buffer (20 mM Tris-HCl, pH 7.5, 100 mM NaCl, 100 mM imidazole) before eluting with elution buffer (20 mM Tris-HCl, pH 7.5, 100 mM NaCl, 300 mM imidazole). The protein was further purified by loading it on a HeparinTrap (GE Healthcare) and eluted with a gradient from 100 mM to 1 M NaCl. Protein-containing fractions were pooled, recombinant His-tagged 3C protease (EMBL PEPcore) was added and the protein was dialyzed overnight against 20 mM HEPES−KOH, pH 7.5, 100 mM NaCl, 2 mM MgCl$_2$, 20 mM imidazole, 0.05% (v/v) 2-mercaptoethanol. The cleaved tag and 3C protease were removed by running the dialysate over another 5 ml HisTrap column and collecting the unbound protein. The protein was concentrated using centrifugal spin concentrators (Millipore) with an MWCO of 10 kDa. As the final purification step SEC was performed on a Superdex 75 16/600 column (GE Healthcare) with 20 mM HEPES−KOH, pH 7.5, 100 mM NaCl, 2 mM MgCl$_2$, 1 mM dithiothreitol (DTT). Fractions with pure protein were pooled and concentrated. All protein purification steps were carried out at 4 °C.

SerRS was purified by resuspending the cell pellet in lysis buffer (20 mM Tris-HCl, pH 7.5, 250 mM NaCl, 20 mM imidazole, 0.05% (v/v) 2-mercaptoethanol, 1 mM PMSF, 1 µg ml$^{-1}$ DNase I, 0.5 µg ml$^{-1}$ RNase and 50 µg ml$^{-1}$ lysozyme) and lysed by microfluidization with a pressure of 275 kPa. The lysate was cleared by centrifugation at 20,000g followed by filtration with a 5 µM filter before loading on a HisTrap HP column (GE Healthcare). The column was washed first with lysis buffer, then with high-salt buffer (20 mM Tris-HCl, pH 7.5, 250 mM NaCl, 1 M KCl, 50 mM imidazole) and low-salt buffer (20 mM Tris-HCl, pH 7.5, 100 mM NaCl, 50 mM imidazole) before eluting with elution buffer (20 mM Tris-HCl, pH 7.5, 100 mM NaCl, 300 mM imidazole). The eluted protein was then applied to a 5 ml Q-trap (GE Healthcare) and eluted with a gradient from 100 mM to 1 M NaCl. Pure protein-containing fractions were pooled and concentrated using centrifugal spin concentrators with an MWCO of 30 kDa (Millipore). Finally, SEC was performed on a Superdex 200 10/300 column with 20 mM HEPES−KOH, pH 7.5, 100 mM NaCl, 2 mM MgCl$_2$, 1 mM DTT. Fractions containing the pure protein were pooled and concentrated. The purified proteins were flash-frozen in liquid nitrogen and stored at −80 °C for further experiments.

### Cryo-EM sample preparation

The DNA sequences coding for SerRS and METTL6 were cloned into a psLIB vector by Gibson Assembly, yielding a fusion construct with the sequence for METTL6 fused directly to the C terminus of SerRS with a C-terminal 3C protease cleavable EGFP tag. The fusion protein was expressed in Hi5 cells. After 72 h of protein expression at 25 °C, the cells were collected by centrifugation, resuspended in lysis buffer (20 mM Tris-HCl, pH 7.5, 100 mM NaCl, 2 mM MgCl$_2$, 0.05% (v/v) 2-mercaptoethanol, 1 mM PMSF) and lysed by sonication. The lysate was

cleared by centrifugation with 20,000g at 4 °C followed by filtration with a 5 µM filter. All subsequent purification steps were carried out at 4 °C.

The lysate was incubated with 1 ml EGFP nanobody resin for 30 minutes. The resin was washed two times with a wash buffer containing 20 mM Tris-HCl, pH 7.5, 100 mM NaCl and 2 mM MgCl$_2$. The SerRS−METTL6 fusion construct was eluted from the resin by cleavage of the EGFP tag using the 3C protease cleavage site: 500 µl wash buffer with 0.08 mg ml$^{-1}$ 3C protease (EMBL PEPcore) was added to the resin and incubated for approximately 2 h. The cleaved protein was removed from the resin, filtered with a Spin-X centrifuge tube filter (Costar) with a pore size of 0.22 µM and concentrated using a centrifugal spin concentrator (Millipore) with an MWCO of 10 kDa. As the final purification step, the protein was purified by SEC on a Superdex 200 3.2/300 column equilibrated in 20 mM HEPES−KOH, pH 7.5, 100 mM NaCl, 2 mM MgCl$_2$, 0.5 mM TCEP. Fractions containing the fusion construct bound to tRNA were frozen with liquid nitrogen and stored at −80 °C until grid preparation.

The sample was diluted to 0.2 mg ml$^{-1}$ with a buffer containing 20 mM HEPES−KOH (pH 7.5), 50 mM NaCl, 2 mM MgCl$_2$, 0.5 mM TCEP and 1 mM sinefungin. For electron microscopy grid preparation, UltrAuFoil grids with an R 1.2/1.3 300 mesh (EMS) were glow-discharged on both sides for 40 seconds with 30 mA at 0.45 bar using a Pelco EasyGlow glow-discharging device. Then, 2 µl of the diluted sample was applied to each side of the grid and the sample was vitrified in liquid ethane using a MARK IV Vitrobot (FEI) with the following settings: 100% humidity, 4 °C, 3 seconds blot time and a blot force of 0.

### Cryo-EM data acquisition, processing and model building

Data were acquired on a Titan Krios transmission electron microscope (FEI) equipped with a 300 kV accelerating voltage field emission gun electron source, a Quantum energy filter (Gatan) and a Gatan K3 camera, at ×130,000 magnification with a pixel size of 0.645 Å per pixel. The microscope was operated using the software SerialEM[67]. Videos were acquired in counting mode with an electron dose of 63.27 e−/Å$^2$ in 40 frames with a defocus range from −0.8 µm to −1.8 µm.

Motion correction and contrast transfer function (CTF) estimation on the collected micrographs were performed using RELION v.3.1 (ref. 68). Particles were picked with the BoxNet_20220403_191253 model in Warp[69], and extracted in RELION in a 550-pixel box. A total of 1,625,946 particles were used as input for data processing using cryoSPARC[70]. After an initial round of two-dimensional (2D) classification to remove bad particles and particles containing only tRNA, the remaining 1,282,076 particles were used for several rounds of ab initio 3D classifications, which resulted in different classes containing dimeric SerRS with one or two tRNA molecules and 0−2 copies of METTL6 bound (Extended Data Figs. 2 and 3 and Table 2). The particles of the 3D class with dimeric SerRS, two tRNAs and one molecule of METTL6 were subjected to several rounds of CTF refinement and Bayesian polishing in RELION v.3.1. The refined particles were re-imported into cryoSPARC to perform a final nonuniform refinement[71]. All reported resolutions were determined using the gold standard FSC method implemented in cryoSPARC, with a cutoff of 0.143.

The highest resolution model was built in Coot using the available crystal structures of tRNA-bound human SerRS (PDB 4RQE)[37] and of METTL6 (this study) as starting templates. The model was refined by several rounds of real-space refinement in Phenix (v.1.16) and manual refinement in Coot (v.0.8.9.2). The other models were built based on the highest resolution model, with minimal refinement. Illustrations were generated with ChimeraX[66], maps shown at a level of 0.3, and PyMOL[72], maps shown at a $\sigma$ level 0.18.

### Protein crystallography

METTL6 was crystallized by vapor diffusion in the hanging drop format at 20 °C at a concentration of 4 mg ml$^{-1}$ supplanted with 2 mM SAH and 5 µM TCEP. A 1 µl portion of protein solution was added to 1 µl of reservoir solution containing 0.1 M Bis-Tris propane, pH 6.5, 0.2 M

sodium sulfate and 20% PEG 3350. After two weeks crystals were soaked for two minutes in a cryoprotectant solution containing 2 mM SAH, 0.1 M Bis-Tris propane, pH 6.5, 0.2 M sodium sulfate, 10% PEG 400 and 30% PEG 3350 and flash-frozen in liquid nitrogen.

A truncated construct of METTL6 consisting of residues 40 to 269 was crystallized at a concentration of 5 mg ml$^{-1}$ supplanted with 1 mM SAH in the sitting drop format platform by adding 100 nl protein solution to 100 nl of a reservoir solution containing 0.1 M Bis-Tris, 0.2 M ammonium sulfate and 25% PEG 3350 at 20 °C. The crystals were collected using the automated CrystalDirect collecting system of the HTX platform[73] and flash-frozen without cryoprotectant.

X-ray diffraction datasets of full-length METTL6 were collected on the MASSIF-1 beamline at the ESRF with an X-ray energy of 12.65 keV (0.9801 Å). For solving the phase problem for METTL6 by single-wavelength anomalous dispersion (SAD) with the anomalous signal from sulfur, a total of 35 X-ray diffraction datasets with an X-ray energy of 6 keV (2.066 Å) were collected on beamline P13 at the PETRA-III storage ring at DESY. X-ray diffraction datasets of METTL6Δ were also obtained on this beamline with an X-ray energy of 12.3 keV (1.008 Å).

All diffraction images were processed using XDS[74]. To solve the phase problem, five of the 6-keV datasets were clustered and merged using BLEND[75] and fed into the CRANK2 pipeline for SAD phasing[75], which resulted in a low-resolution (4.42 Å) structure of METTL6. This low-resolution structure was used to obtain phases for the other datasets by molecular replacement using Phenix-PHASER[76]. For all models, several rounds of model building using Coot[77] and data refinement using Phenix-REFINE[78] were performed. Ramachandran statistics for the METTL6 (full length) are 97.42% favored, 2.58% allowed and 0% disallowed, and for the METTL6 (40–269) are 95.18% favored, 4.82% allowed and 0% disallowed. Data collection and refinement statistics are shown in Table 1.

## tRNA in vitro transcription
The DNA sequences coding for a T7 promoter, human tRNA$^{Ser}_{UGA}$ or tRNA$^{Thr}_{CGU}$, respectively, directly followed by a BstN1 cleavage site, were cloned into a pUC19 vector. The vector was linearized using BstN1 and used as a template for runoff transcription with T7 polymerase. A 1 mg portion of linearized DNA was transcribed in vitro in a reaction containing 40 mM Tris-HCl, pH 8.0, 30 mM MgCl$_2$, 5 mM DTT, 1 mM spermidine, 0.01% (w/w) Triton X100, 4 mM of each NTP and 50 µg ml$^{-1}$ recombinant T7 polymerase (EMBL PEPcore), which was incubated for 16 h at 37 °C. The produced RNA was precipitated with isopropanol, purified via urea–PAGE, desalted using a Pierce dextran desalting column (ThermoFisher) and concentrated using a centrifugal spin concentrator (Millipore) with a MWCO of 10 kDa. The tRNA was refolded by heating it for 2 minutes to 95 °C before adding MgCl$_2$ to a final concentration of 2 mM and quickly placing it on ice.

## Thermal shift assay
METTL6 was diluted to a final concentration of 20 µM in a buffer containing 20 mM HEPES–KOH (pH 7.5), 100 mM NaCl, 2 mM MgCl$_2$, 1 mM DTT, 5× SYPRO Orange (Invitrogen) and 1 mM of cofactor (SAM, SAH or sinefungin). Fluorescence was measured in a real-time PCR machine (Stratagene Mx3005P) while subjected to a temperature gradient from 24.6 °C to 95°°C in steps of 1 °C and 1 min.

## Fluorescence polarization assay
The in-vitro-transcribed tRNA$^{Ser}_{UGA}$ and tRNA$^{Thr}_{CGU}$ were labeled on the 3′ end with fluorescein. The DNA and RNA oligomers (labeled on the 3′ end with 6-carboxyfluorescein) were purchased from biomers.net. All nucleic acids were refolded as described above. The labeled tRNAs with a concentration of 25 nM were incubated with METTL6 of different concentrations in a total volume of 20 µl for 15 minutes at room temperature in a 384-well flat black microplate (Greiner) in a buffer of 20 mM HEPES–KOH (pH 7.5), 50 mM NaCl, 2 mM MgCl$_2$, 0.1% Tween, 1 mM DTT, 0.5 U µl$^{-1}$ RNasin and 0.5 mM sinefungin. Fluorescence

polarization was measured at a temperature of 25 °C with a ClarioStar plate reader (BMG Labtech) with an excitation wavelength of 460 nm and an emission wavelength of 515 nm. Data analysis was performed using GraphPad Prism using the single site binding fit.

## Size-exclusion complex formation assay
METTL6, SerRS and tRNA$^{Ser}_{UGA}$ (each at a concentration of 30 µM; METTL6 supplemented with 1 mM sinefungin to stabilize the complex) were incubated for 10 minutes on ice before subjecting them to analytical SEC using a Superdex 200 Increase 3.2/300 column (GE Healthcare) equilibrated in a buffer composed of 20 mM HEPES–KOH, pH 7.5, 50 mM NaCl, 2 mM MgCl$_2$, 0.5 mM TCEP. Fractions were analyzed via SDS–PAGE stained with Coomassie blue and urea–PAGE stained with methylene blue for protein and RNA content, respectively.

## Methyltransferase assay
Methyltransferase assays were performed in 6 mM HEPES–KOH (pH 7.9), 0.4 mM EDTA, 10 mM DTT, 80 mM KCl, 1.5 mM MgCl$_2$, RNasin (40 U µl$^{-1}$) (Promega) and 1.6% glycerol in a total volume of 200 µl. Proteins (METTL6, SerRS) were used in 0.05 µM final concentration. As substrate, 0.4 µM of in-vitro-transcribed tRNA(Ser) or tRNA(Thr), or 5 µg of total RNA extracted from HeLa cells, was added. In addition, 4 µM of serine (Sigma) and 2 mM of ATP were added where indicated. A 1 µl portion of $^3$H-labeled SAM (1 mCi ml$^{-1}$; PerkinElmer) was added to the mixture and then incubated at 30 °C for 1 h with gentle shaking. Then, another 1 µl of $^3$H-SAM was added and the assay continued at room temperature (22 °C) overnight followed by column purification of the RNA with the Zymo Research RNA Miniprep kit (used according to the manufacturer's instructions). After elution of RNA from the columns in 150 µl nuclease-free H$_2$O, 500 µl of AquaLight liquid scintillation counter cocktail (Hidex) was added and tritium incorporation analyzed by scintillation counting with a Triathler Counter (Hidex). All data of in vitro methyltransferase assays are shown as blank-subtracted mean of three scintillation counts and three experimental replicates that were carried out for each experiment.

## Sample preparation for LC–MS/MS and detection of modified nucleosides
The *Trichoplusia ni* tRNA that copurified with the SerRS–METTL6 fusion construct was isolated via chloroform–phenol extraction and precipitated with ethanol and sodium acetate. Around 300 ng of the tRNA was digested in an aqueous digestion mix (30 µl) to single nucleosides by using 2 U alkaline phosphatase, 0.2 U phosphodiesterase I (VWR) and 2 U benzonase in Tris (pH 8, 5 mM) and MgCl$_2$ (1 mM) containing buffer. Furthermore, 0.5 µg tetrahydrouridine (Merck), 1 µM butylated hydroxytoluene and 0.1 µg pentostatin were added. After incubation for 2 h at 37 °C, 20 µl of LC–MS buffer A (QQQ) was added to the mixture. A stable isotope labeled SILIS (gen² (ref. 79)), was added to each replicate and calibration solution of synthetic standards before injection into the QQQ MS.

## LC–MS/MS of nucleosides
Modified nucleosides were identified and quantified using mass spectrometry. An Agilent 1290 Infinity II equipped with a diode-array detector combined with an Agilent Technologies G6470A Triple Quad system and electrospray ionization (ESI) mass spectrometer (Agilent Jetstream) was used.

Nucleosides were separated using a Synergi Fusion-RP column (Synergi 2.5 µm Fusion-RP 100 Å, 150 mm × 2.0 mm, Phenomenex). LC buffer consisting of 5 mM NH$_4$OAc, pH 5.3 (buffer A) and pure acetonitrile (buffer B) was used. The gradient started with 100% buffer A for 1 min, followed by an increase to 10% buffer B over a period of 4 min. Buffer B was then increased to 40% over 2 min and maintained for 1 min before switching back to 100% buffer A over a period of 0.5 min and re-equilibrating the column for 2.5 min. The total time was 11 min and the flow rate was 0.35 ml min$^{-1}$ at a column temperature of 35 °C.

# Article

An ESI source was used for ionization of the nucleosides (Agilent Jetstream). The gas temperature ($N_2$) was 230 °C with a flow rate of 6 l min⁻¹. Sheath gas temperature was 400 °C with a flow rate of 12 l min⁻¹. Capillary voltage was 2,500 V, skimmer voltage was 15 V, nozzle voltage was 0 V and nebulizer pressure was 40 psi. The cell accelerator voltage was 5 V. For nucleoside modification screening, MS2Scan ($m/z$ 250–500) was used. For quantification, a DMRM and positive-ion mode was used (Supplementary Table 2).

## Data analysis of nucleosides

For calibration, synthetic nucleosides were weighed and dissolved in water to a stock concentration of 1–10 mM. Calibration solutions ranged from 0.0125 pmol to 100 pmol for each canonical nucleoside, and from 0.00625 pmol to 5 pmol for each modified nucleoside. Analogous to the samples, 1 µl SILIS (10×) was co-injected with each calibration. The calibration curve and the corresponding evaluation of the samples were performed using Agilent's quantitative MassHunter software. All modification abundances were normalized to the amount of RNA injected using the sum of all canonicals.

## Reporting summary

Further information on research design is available in the Nature Portfolio Reporting Summary linked to this article.

## Data availability

The model coordinates are deposited in the PDB with accession codes 8P7B for the 1:2:2 METTL6–SerRS–tRNA complex, 8P7C for the 2:2:2 complex and 8P7D for the 1:2:1 complex. EMDB entries, in the same order, are EMD-17528, EMD-17529 and EMD-17530. Maps of SerRS–tRNA and tRNA 3D classes were side products of the data processing and have been deposited under EMD-17531 and EMD-17532. All micrographs of the dataset are deposited at EMPIAR-11578. The crystal structures of METTL6 are deposited in the PDB (8OWX and 8OWY). Source data are provided with this paper.

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

## Acknowledgements

We acknowledge M. Pelosse for support in using the Eukaryotic Expression Facility at EMBL Grenoble, and S. Schneider for support in using the EM Facility at EMBL Grenoble. This work benefited from access to the cryo-EM platform of the Structural and Computational Biology Unit at EMBL Heidelberg and we thank F. Weis for his assistance with the data collection. We acknowledge the European Synchrotron Radiation Facility and the Deutsches Elektronen-Synchrotron for provision of synchrotron radiation facilities and we would like to thank the staff of the ESRF and EMBL Grenoble at beamline MASSIF-1, and DESY and EMBL Hamburg at beamline P13, respectively, for assistance and support. We acknowledge the HTX Team for crystallization setups, automated crystal collection and thermostability assays. This work used the platforms of the Grenoble Instruct-ERIC Centre (ISBG; UAR 3518 CNRS-CEA-UGA-EMBL) within the Grenoble Partnership for Structural Biology (PSB), supported by FRISBI (ANR-10-INBS-0005-02) and GRAL, financed within the University Grenoble Alpes graduate school (Écoles Universitaires de Recherche) CBH-EUR-GS (ANR-17-EURE-0003). The laboratory of E.K. is supported through EMBL core funding and a grant from the French Agence Nationale de la Recherche to E.K (ANR-20-CE11-0016-01). Work in the R.S. laboratory was supported by the DFG through SFB 1064 (Project-ID 213249687) and the Helmholtz Gesellschaft. Both R.S. and S.K. are supported through SFB 1309 (Project-ID 325871075). We thank the Kowalinski laboratory members for discussions and comments throughout the course of the project.

## Author contributions

E.K. conceptualized and led the study. P.T., L.T. and L.G.D. conducted protein purifications, RNA in vitro transcription, SEC experiments, FP and cryo-EM sample preparation. S.B. assisted in X-ray data collection, phasing and data processing. P.R.L. and K.A. conducted activity assays in the laboratory of R.S. S.K. conducted nucleoside modification mass spectrometry. P.T. and L.G.D. collected and processed cryo-EM data. P.T., L.G.D. and E.K. built the models and interpreted the data. E.K. wrote the manuscript supported by L.G.D., P.T. and R.S.

## Funding

## Competing interests

The authors declare no competing interests.

## Additional information

**Extended data** is available for this paper at https://doi.org/10.1038/s41594-024-01341-3.

**Correspondence and requests for materials** should be addressed to Eva Kowalinski.

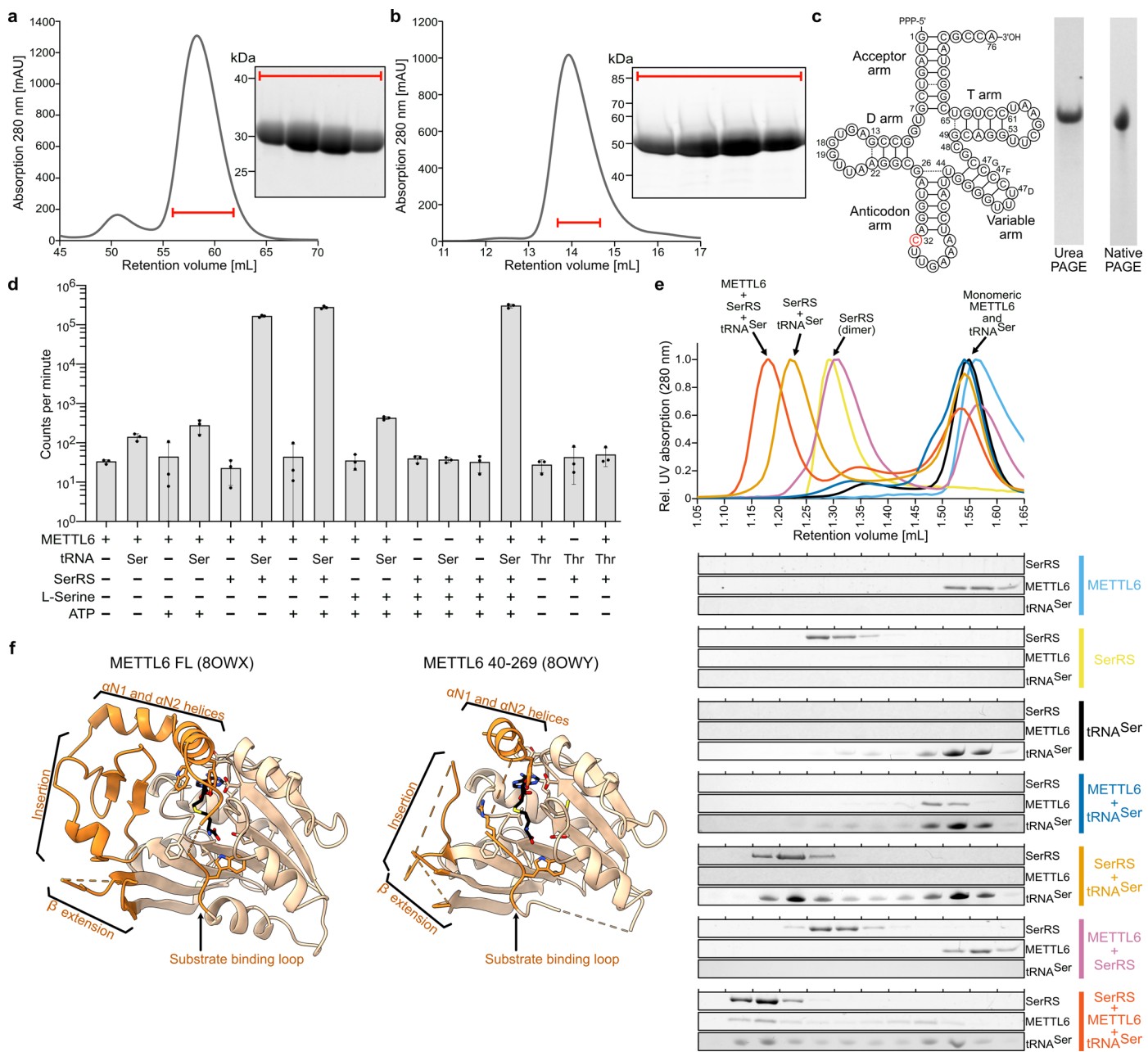

**Extended Data Fig. 1 | Protein/RNA purification and cryo-EM sample preparation. a** Size–exclusion chromatogram and SDS-PAGE analysis of the last step of the purification of full-length METTL6 (column: Superdex 75 16/600); representative chromatogramme and gel from more than three replicates. **b** Size-exclusion chromatogram and SDS-PAGE analysis of the last step of the purification of full-length SerRS (Column: Superdex 200 10/300); representative chromatogramme and gel from more than three replicates. **c** Sequence (left) and quality control of *in vitro* transcribed tRNA$^{Ser}_{UGA}$ on denaturing urea PAGE (left) and native PAGE (right); representative gels from more than three replicates.

**d** *In vitro* methylation activity measurements of METTL6 in the presence of SerRS, tRNA$^{Ser}_{UGA}$ and/or L-Serine and ATP. Data is represented as the blank-subtracted mean and standard deviation of independent replicates (n = 3). **e** Complex formation of SerRS, tRNA$^{Ser}_{UGA}$ and METTL6 on analytical size-exclusion chromatography (Column: Superdex 200 3.2/300); representative chromatogramme and gel from one or more replicates. Fractions were analysed for protein and RNA contents using SDS-PAGE and urea PAGE. **f** Cartoon representation of crystal structures from this study: apo-METTL6 full-length (pdb:8OWX) and apo-METTL6 40–269 (pdb 8OWY) in complex with SAH.

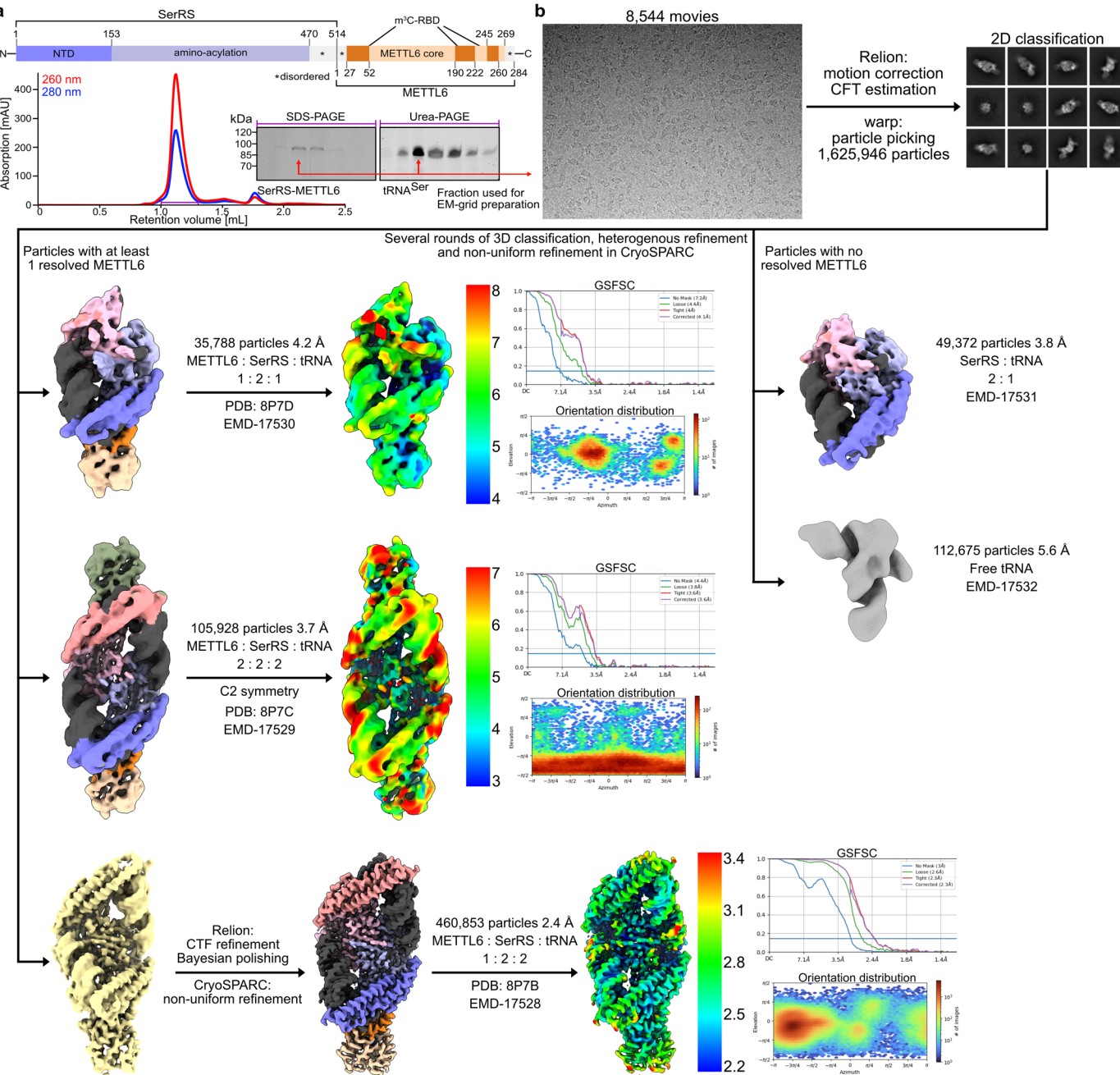

**Extended Data Fig. 2 | Cryo-EM data processing scheme. a** Cryo-EM sample preparation with a schematic of the SerRS-METTL6 fusion polypeptide. Size-exclusion chromatogram and SDS-PAGE and urea PAGEs. Representative chromatogramme and gels from more than three replicates. The fraction marked with a red arrow was subjected to cryo-EM grid vitrification. **b** Cryo-EM data processing scheme. A representative micrograph and representative 2D classes are shown. The final resolution estimates were calculated using the gold standard FSC method implemented in CryoSPARC. Colouring of the structure by components as in main Fig. 1.

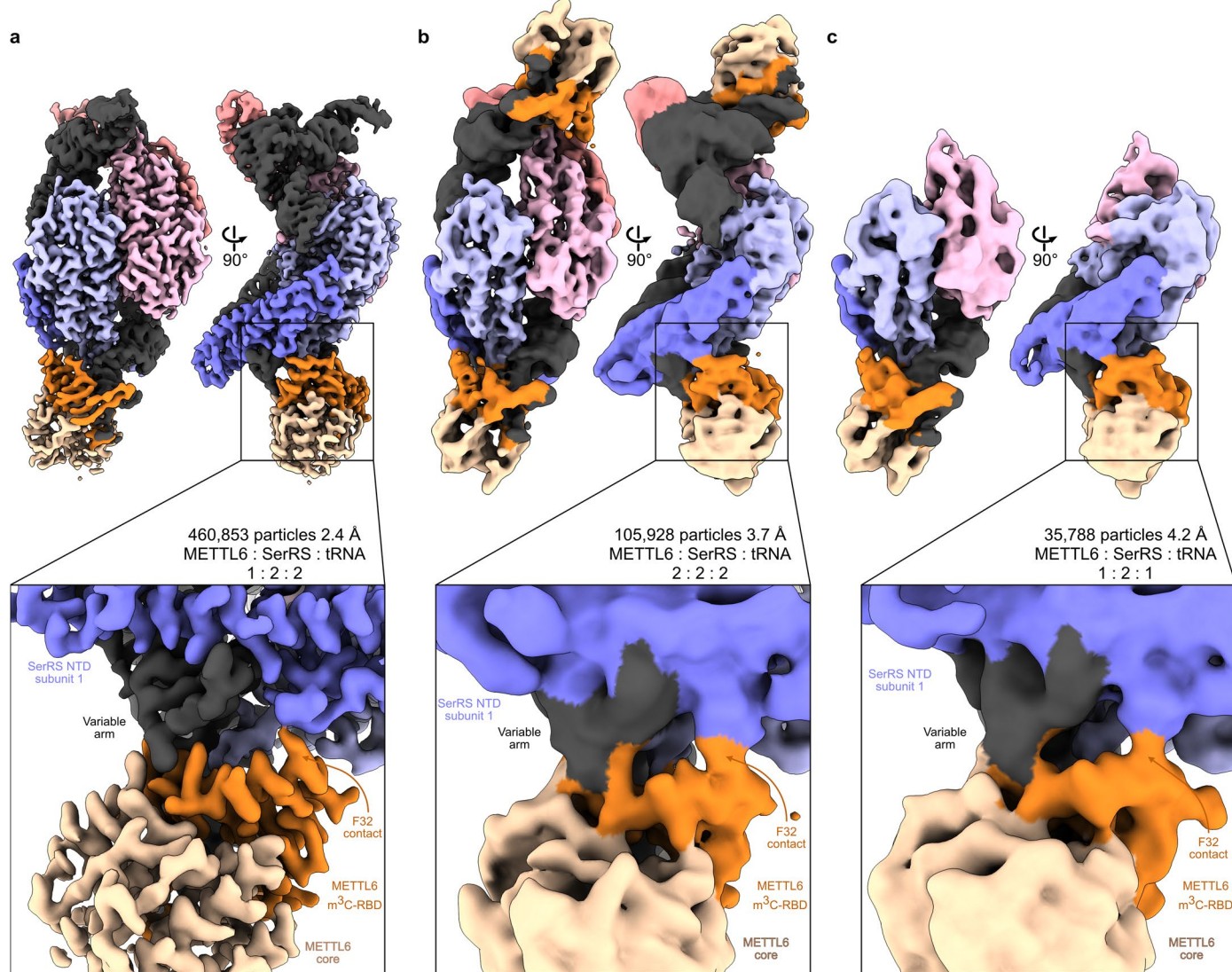

460,853 particles 2.4 Å
METTL6 : SerRS : tRNA
1 : 2 : 2

105,928 particles 3.7 Å
METTL6 : SerRS : tRNA
2 : 2 : 2

35,788 particles 4.2 Å
METTL6 : SerRS : tRNA
1 : 2 : 1

**Extended Data Fig. 3 | Comparison of the METTL6-SerRS-tRNASer interfaces between the reconstructed 3D classes.** Final EM map of **a** 1:2:2 complex at 2.4 Å resolution and represented at level 0.3. **b** 2:2:2 complex at 3.7 Å resolution and represented at level 0.162. **c** 1:2:1 complex at 4.2 Å resolution and represented at level 0.168. All are coloured as in Fig. 1c, with the methyltransferase core of METTL6 coloured in light orange and the RNA binding domain (m³C-RBD) in dark orange. The N-terminal domain (NTD) of SerRS is coloured in dark purple and the aminoacylation domain (AD) in light purple. The second SerRS protomer NTD is coloured in salmon and the AD in light pink.; tRNA is coloured in black.

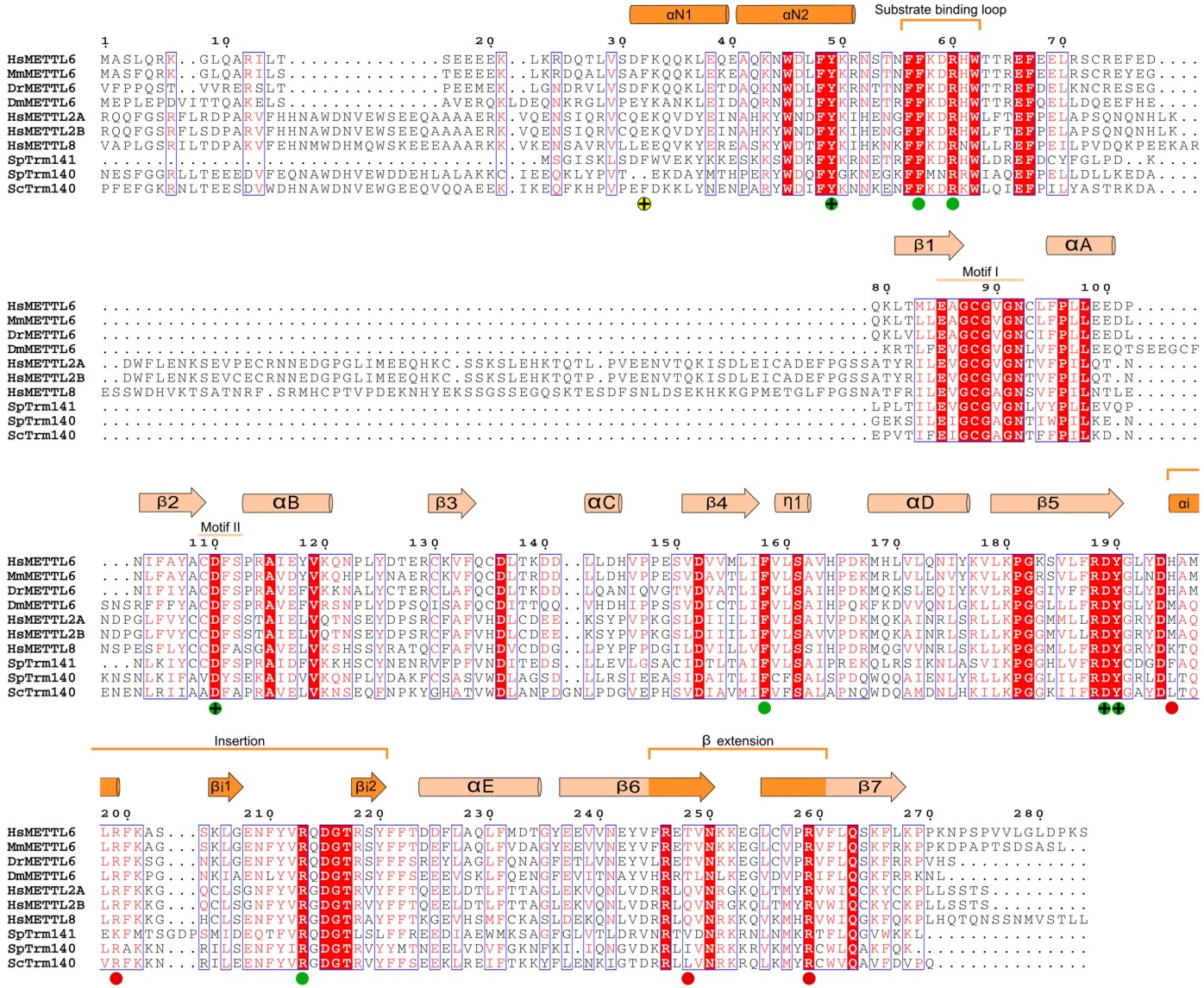

**Extended Data Fig. 4 | Multiple sequence alignment of m3C RNA methyltransferases.** White letters on red background signify 100% of conservation, blue boxes indicate >70% residue similarity with the conserved residues in red. A secondary structure representation of human METTL6 from our cryo-EM model is shown at the top. Residues involved in the coordination of $C_{32}$ are annotated with a green circle. Phenylalanine F32 in the METTL6-SerRS interface is annotated with a yellow circle. Residues involved in the coordination of $i^6A_{37}$ are annotated with a red circle. Single-point mutants analysed in this study are additionally annotated with a '+' sign.

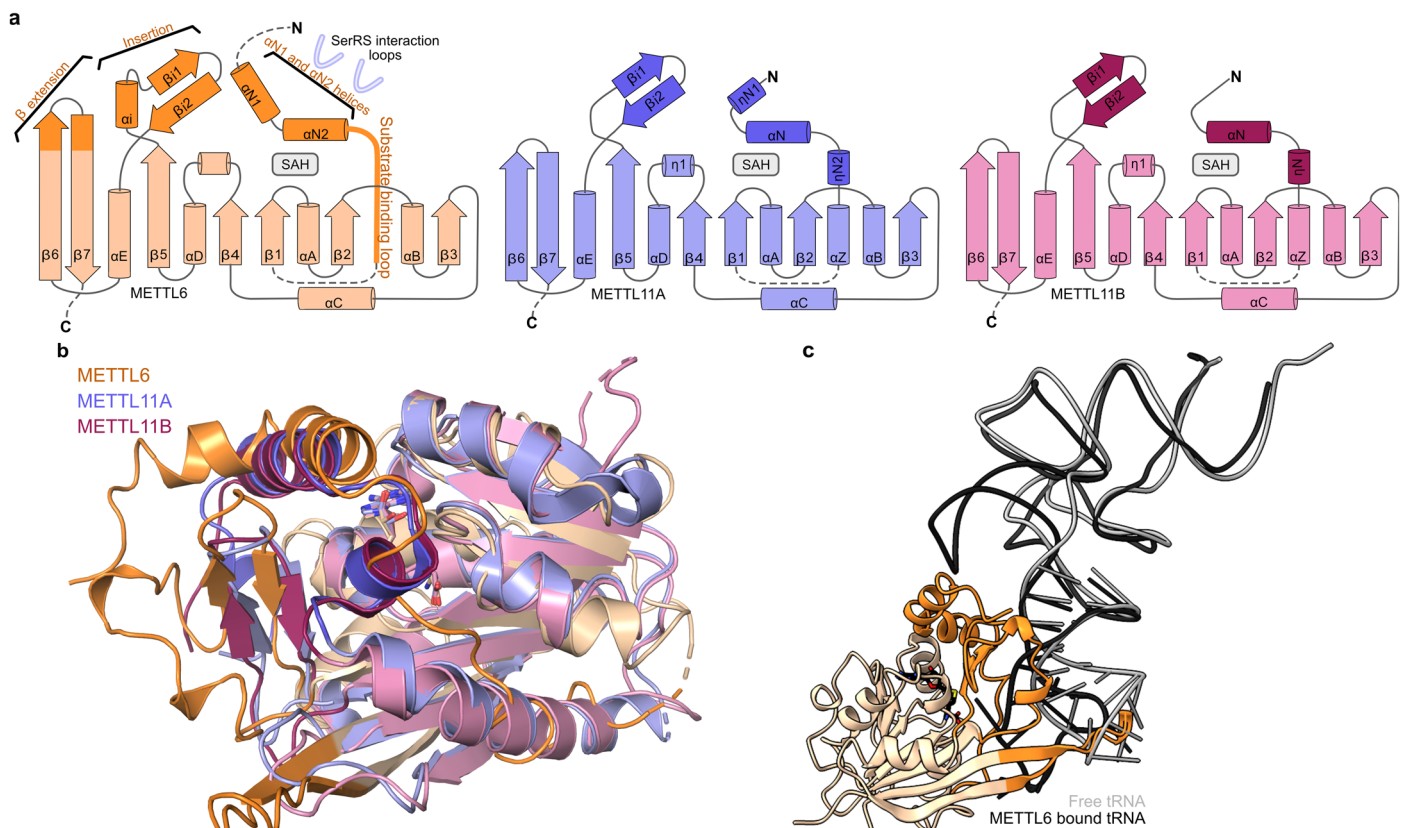

**Extended Data Fig. 5 | Comparison with METTL11. a** Secondary structure diagrams of METTL6 in comparison with METTL11A (PDB:5UBB) and METTL11B (PDB:6DUB). Dotted lines represent unmodelled parts of the experimental structures. **b** Superposition of the model of METTL6 from the METTL6-SerRS-

tRNA$^{Ser}$ complex with METTL11A (PDB:5UBB, rmsd = 1.137 Å) and METTL11B (PDB:6DUB, rmsd = 1.102 Å) **c** Distortion of the anticodon arm in the METTL6-SerRS-tRNA$^{Ser}$ complex compared to free tRNA (PDB:1ehz, grey) (rmsd = 1.124 Å).

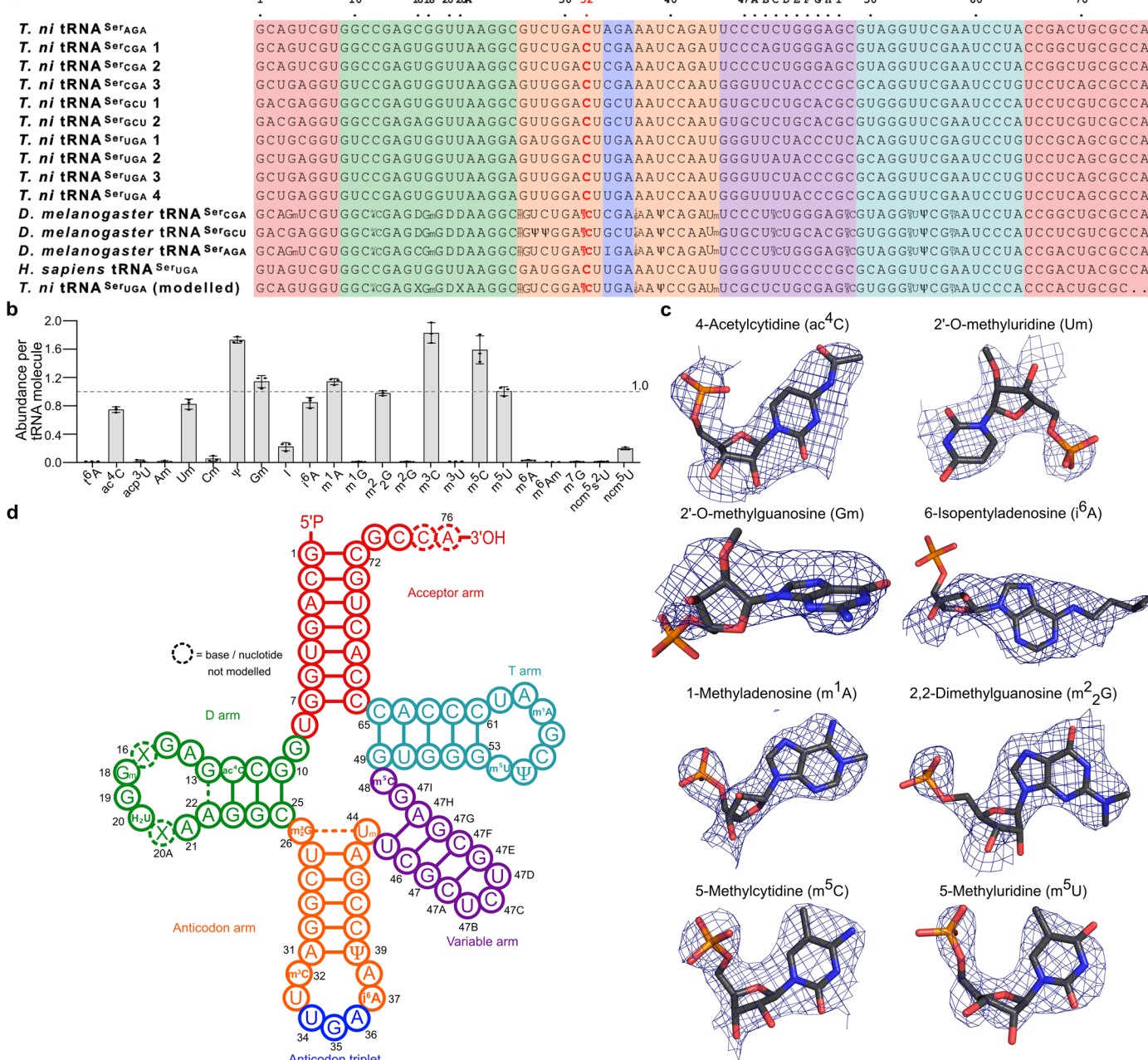

**Extended Data Fig. 6 | Modifications in the co-purified tRNA$^{Ser}$ from *T. ni* insect cell expression. a** Sequence alignment of the ten tRNA$^{Ser}$ isoacceptors present in expression host *T. ni*, the three tRNA$^{Ser}$ from *D. melanogaster* with mapped modifications, *H. sapiens* tRNA$^{Ser}_{UGA}$ and the consensus sequence we used for our model. The background colours represent the tRNA secondary structure features. **b** Relative abundance of modified nucleotides per tRNA molecule as determined by mass spectrometry from three biological replicates. Data are represented as mean values ± standard deviation. See also Extended Table 4. **c** Example of the cryo-EM map around modified nucleotides at positions specified in panel d. Coulomb density is shown around the model. Stick model with heteroatom colour scheme. **d** Cloverleaf representation of the tRNA$^{Ser}$ sequence in the model with colours as in panel a.

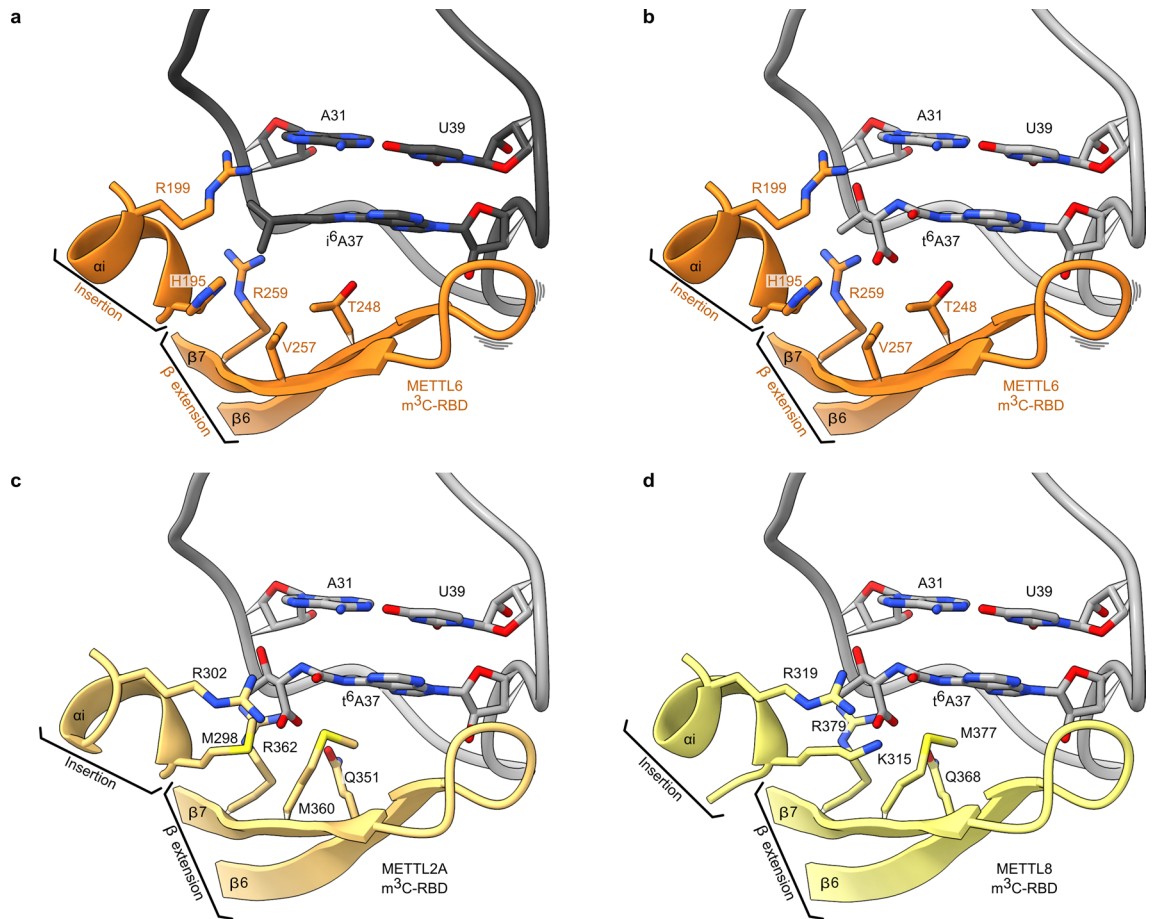

**Extended Data Fig. 7 | Modeling of the coordination of possible A37 adenosine modifications. a** Experimental structure of METTL6-SerRS-tRNA$^{Ser}$ with N$^6$-isopentenyladenosine i$^6$A$_{37}$. **b** The same experimental protein structure of METTL6 with modelled N$^6$-threonylcarbamoyladenosine t$^6$A$_{37}$, that could putatively form several hydrogen bonds with the protein. **c** Alphafold DB model (B3KW44) of METTL2A with t$^6$A. **d** Alphafold DB model (Q96IZ6) of METTL8 with t$^6$A.

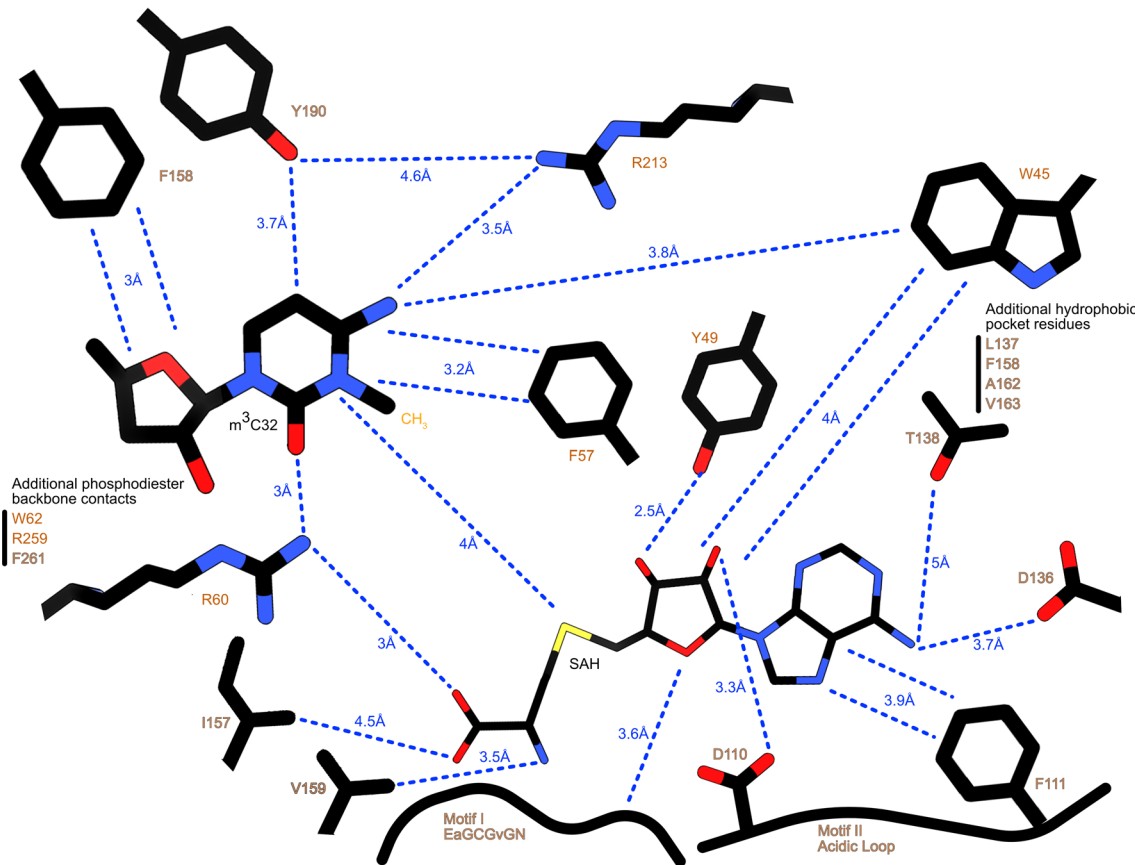

**Extended Data Fig. 8 | Two-dimensional schematic of the METTL6 active site.** METTL6 side chains, active site motifs, m³C₃₂ and SAH are sketched in black and interatomic distances are indicated by dotted lines. Additional interacting residues are indicated as text.

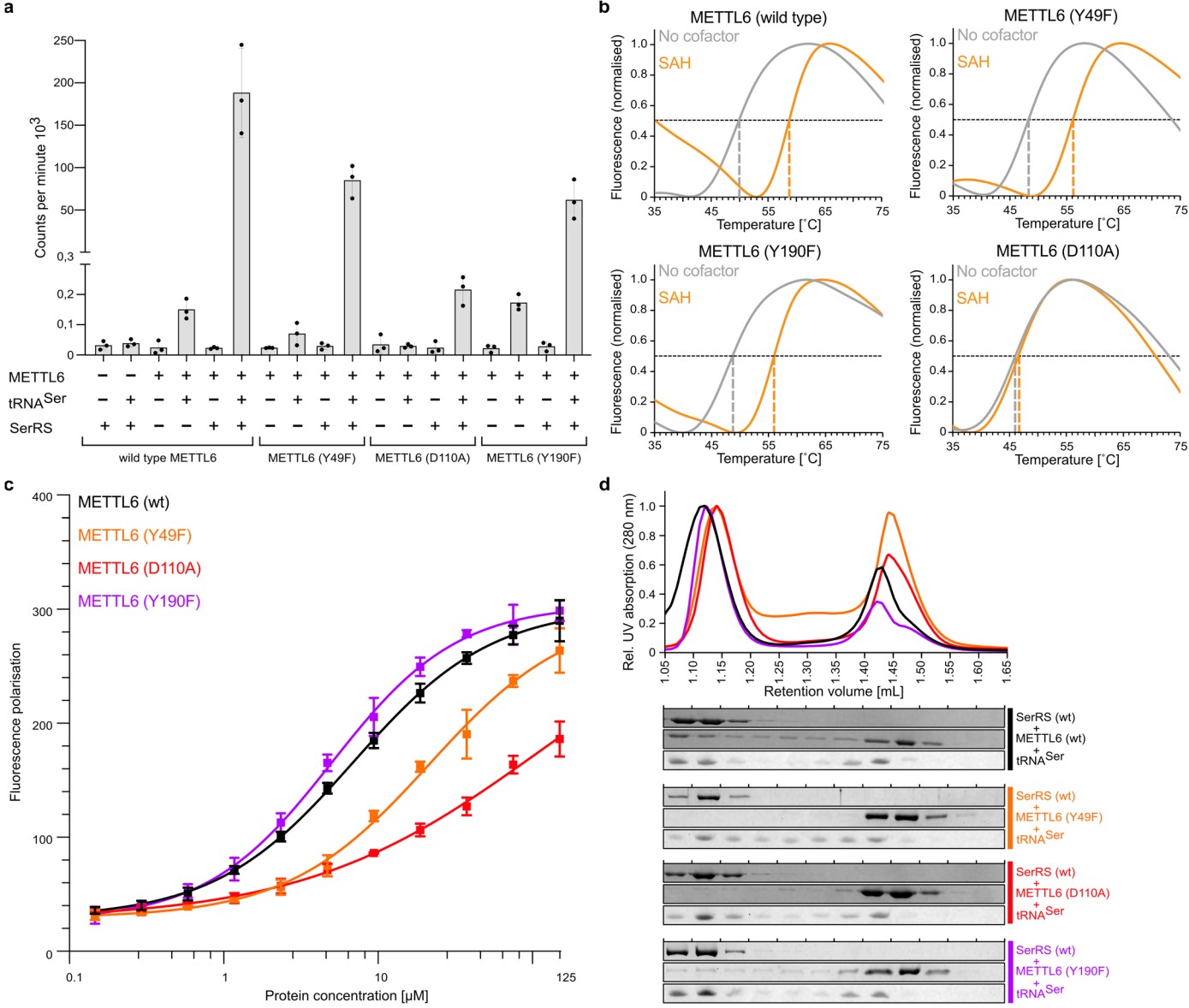

**Extended Data Fig. 9 | Characterization of METTL6 mutants. a** *In vitro* methylation activity measurements of the different METTL6 mutants. Data is represented as the blank-subtracted mean and standard deviation of independent replicates (n = 3). **b** Thermal shift assay of METTL6 wild type and mutants, in the presence or absence of SAH. An increase in the melting temperature indicates SAH binding. A representative melting curve of 2 replicates is shown. **c** Interaction of the different METTL6 mutants with

tRNA$^{Ser}_{UGA}$ measured by fluorescence polarisation. Data is represented as the mean and standard deviation of independent replicates (n = 3), and the sigmoidal fit of the curve. **d** Complex formation of SerRS mutants, METTL6 mutants and tRNA$^{Ser}_{UGA}$ qualitatively analysed via analytical size-exclusion chromatography (Column: Superdex 200 3.2/300); representative chromatogramme and gel from one or more replicates. Fractions were analysed for protein and RNA contents using SDS-PAGE and urea PAGE.

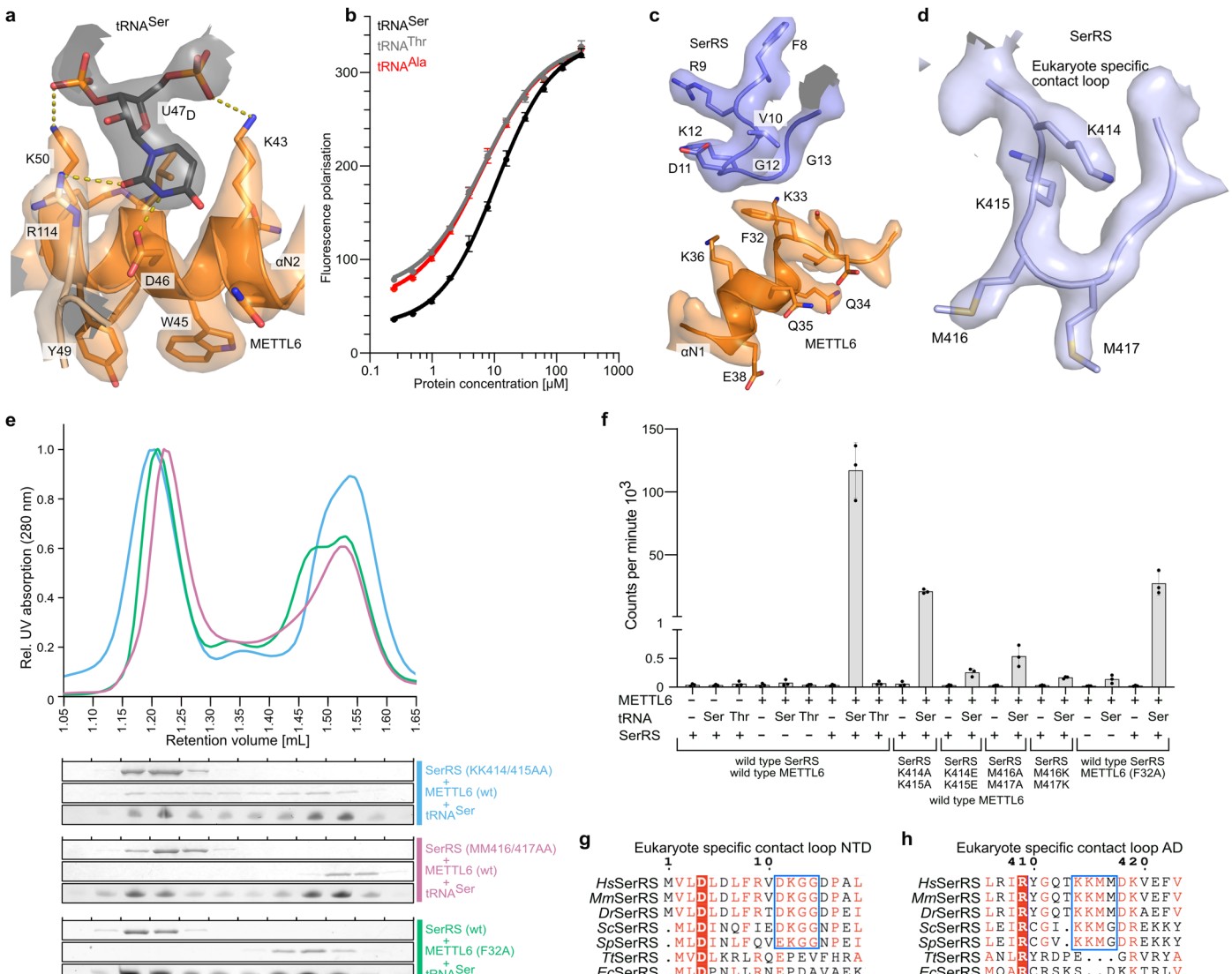

**Extended Data Fig. 10 | Details of METTL6 - SerRS interaction. a** Close-up view of METTL6 binding of U47$_D$ with the corresponding section of the cryo-EM map. **b** Interaction of METTL6 with different tRNAs measured by fluorescence polarisation. Data is represented as the mean and standard deviation of independent replicates (n = 3), and the sigmoidal fit of the curve. **c** Close-up view of the interaction between F32 of METTL6 and the N-terminal domain of SerRS with the corresponding section of the cryo-EM map. **d** Close-up view of the KKMM loop of SerRS with the corresponding section of the cryo-EM

map. **e** Complex formation of SerRS mutants, METTL6 mutants and tRNA$^{Ser}_{UGA}$ qualitatively analysed via analytical size-exclusion chromatography (Column: Superdex 200 3.2/300); representative chromatogramme and gel from one or more replicates. Fractions were analysed for protein and RNA contents using SDS-PAGE and urea PAGE. **f** *In vitro* methylation activity of METTL6 and SerRS mutants on tRNA$^{Ser}$ and tRNA$^{Thr}$. Data is represented as the blank-subtracted mean and standard deviation of independent replicates (n = 3). **g and h** Multiple sequence alignment of SerRS sequences in the METTL6 contact loops.

# Reporting Summary

## Statistics

For all statistical analyses, confirm that the following items are present in the figure legend, table legend, main text, or Methods section.

| n/a | Confirmed | |
|---|---|---|
| ☐ | ☒ | The exact sample size (*n*) for each experimental group/condition, given as a discrete number and unit of measurement |
| ☐ | ☒ | A statement on whether measurements were taken from distinct samples or whether the same sample was measured repeatedly |
| ☒ | ☐ | The statistical test(s) used AND whether they are one- or two-sided<br>*Only common tests should be described solely by name; describe more complex techniques in the Methods section.* |
| ☒ | ☐ | A description of all covariates tested |
| ☒ | ☐ | A description of any assumptions or corrections, such as tests of normality and adjustment for multiple comparisons |
| ☐ | ☒ | A full description of the statistical parameters including central tendency (e.g. means) or other basic estimates (e.g. regression coefficient) AND variation (e.g. standard deviation) or associated estimates of uncertainty (e.g. confidence intervals) |
| ☒ | ☐ | For null hypothesis testing, the test statistic (e.g. *F*, *t*, *r*) with confidence intervals, effect sizes, degrees of freedom and *P* value noted<br>*Give P values as exact values whenever suitable.* |
| ☒ | ☐ | For Bayesian analysis, information on the choice of priors and Markov chain Monte Carlo settings |
| ☒ | ☐ | For hierarchical and complex designs, identification of the appropriate level for tests and full reporting of outcomes |
| ☒ | ☐ | Estimates of effect sizes (e.g. Cohen's *d*, Pearson's *r*), indicating how they were calculated |

*Our web collection on statistics for biologists contains articles on many of the points above.*

## Software and code

Policy information about availability of computer code

| Data collection | n.a |
|---|---|
| Data analysis | n.a |

For manuscripts utilizing custom algorithms or software that are central to the research but not yet described in published literature, software must be made available to editors and reviewers. We strongly encourage code deposition in a community repository (e.g. GitHub). See the Nature Portfolio guidelines for submitting code & software for further information.

## Data

Policy information about availability of data

All manuscripts must include a data availability statement. This statement should provide the following information, where applicable:

- Accession codes, unique identifiers, or web links for publicly available datasets
- A description of any restrictions on data availability
- For clinical datasets or third party data, please ensure that the statement adheres to our policy

The cryo-EM maps and their derived models for the deposited in the PDB: accession code 8P7B for the 1:2:2 METTL6:SerRS:tRNA stochiometry, 8P7C for the 2:2:2 complex and 8P7D for the 1:2:1 complex. EMDB entries in the same order are EMD-17528, EMD-17529 and EMD-17530. Maps of SerRS-tRNA and tRNA 3D-classes were side products of the data processing and have been deposited under EMD-17531 and EMD-17532. All micrographs of the dataset are deposited at EMPIAR-11578. The crystal structures of METTL6 are deposited in the PDB (8OWX and 8OWY). Source data are provided in addition to this paper.

# Research involving human participants, their data, or biological material

Policy information about studies with human participants or human data. See also policy information about sex, gender (identity/presentation), and sexual orientation and race, ethnicity and racism.

| | |
|---|---|
| Reporting on sex and gender | n.a. |
| Reporting on race, ethnicity, or other socially relevant groupings | n.a. |
| Population characteristics | n.a. |
| Recruitment | n.a. |
| Ethics oversight | *Identify the organization(s) that approved the study protocol.* |

Note that full information on the approval of the study protocol must also be provided in the manuscript.

# Field-specific reporting

Please select the one below that is the best fit for your research. If you are not sure, read the appropriate sections before making your selection.

☒ Life sciences  ☐ Behavioural & social sciences  ☐ Ecological, evolutionary & environmental sciences

For a reference copy of the document with all sections, see nature.com/documents/nr-reporting-summary-flat.pdf

# Life sciences study design

All studies must disclose on these points even when the disclosure is negative.

| | |
|---|---|
| Sample size | stated in figure legends and methods |
| Data exclusions | n.a. |
| Replication | stated in figure legends and methods |
| Randomization | n.a. |
| Blinding | n.a. |

# Reporting for specific materials, systems and methods

We require information from authors about some types of materials, experimental systems and methods used in many studies. Here, indicate whether each material, system or method listed is relevant to your study. If you are not sure if a list item applies to your research, read the appropriate section before selecting a response.

## Materials & experimental systems

| n/a | Involved in the study |
|---|---|
| ☒ ☐ | Antibodies |
| ☐ ☒ | Eukaryotic cell lines |
| ☒ ☐ | Palaeontology and archaeology |
| ☒ ☐ | Animals and other organisms |
| ☒ ☐ | Clinical data |
| ☒ ☐ | Dual use research of concern |
| ☒ ☐ | Plants |

## Methods

| n/a | Involved in the study |
|---|---|
| ☒ ☐ | ChIP-seq |
| ☒ ☐ | Flow cytometry |
| ☒ ☐ | MRI-based neuroimaging |

# Eukaryotic cell lines

Policy information about cell lines and Sex and Gender in Research

| | |
|---|---|
| Cell line source(s) | T. ni. |

| | |
|---|---|
| Authentication | commonly used insect cell expression cell line for expression |
| Mycoplasma contamination | n.a. |
| Commonly misidentified lines<br>(See ICLAC register) | *Name any commonly misidentified cell lines used in the study and provide a rationale for their use.* |

