## [Peer Review File · Nature Structural & Molecular Biology]

Peer Review Information

Manuscript Title: Structural basis of tRNA recognition by the m³C-RNA-methyltransferase METTL6 in complex with SerRs seryl-tRNA synthetase

Corresponding author name(s): Eva Kowalinski

Reviewer Comments & Decisions:

Decision Letter, initial version:

Message: 8th Sep 2023

Dear Dr. Kowalinski,

Thank you again for submitting your manuscript "Structural basis of tRNA recognition by the m³C-RNA-methyltransferase METTL6 in complex with SerRS seryl-tRNA synthetase". I sincerely apologize for the delay in responding due to absences in the editorial team. Nevertheless, we now have comments (below) from the 2 reviewers who evaluated your paper. In light of those reports, we remain interested in your study and would like to see your response to the comments of the referees, in the form of a revised manuscript.

You will see that while both reviewers appreciate the results, Reviewer #1 raises some concerns about the relevant stoichiometry of the complex, given the use of a fusion protein in structural determination, and requests more detailed description of the intermolecular interactions given the high resolution obtained. Please be sure to address/respond to all concerns of the referees in full in a point-by-point response and highlight all changes in the revised manuscript text file. If you have comments that are intended for editors only, please include those in a separate cover letter.

We expect to see your revised manuscript within 6 weeks. If you cannot send it within this time, please contact us to discuss an extension; we would still consider your revision, provided that no similar work has been accepted for publication at NSMB or published elsewhere.

Reporting Summary:

When submitting the revised version of your manuscript, please pay close attention to our [Digital Image Integrity Guidelines](https://www.nature.com/nature-portfolio/editorial-policies/image-integrity). and to the following points below:

Please note that all key data shown in the main figures as cropped gels or blots should be presented in uncropped form, with molecular weight markers. These data can be aggregated into a single supplementary figure item. While these data can be displayed in a relatively informal style, they must refer back to the relevant figures. These data should be submitted with the final revision, as source data, prior to acceptance, but you may want to start putting it together at this point.

SOURCE DATA: we request that authors provide, in tabular form, the data underlying the graphical representations used in figures. This is to further increase transparency in data reporting, as detailed in this editorial (<http://www.nature.com/nsmb/journal/v22/n10/full/nsmb.3110.html>). Spreadsheets can be submitted in excel format. Only one (1) file per figure is permitted; thus, for multi-paneled figures, the source data for each panel should be clearly labeled in the Excel file; alternately the data can be provided as multiple, clearly labeled sheets in an Excel file. When submitting files, the title field should indicate which figure the source data pertains to. We encourage our authors to provide source data at the revision stage, so that they are part of the peer-review process.

Data availability: this journal strongly supports public availability of data. All data used in accepted papers should be available via a public data repository, or alternatively, as Supplementary Information. If data can only be shared on request, please explain why in your Data Availability Statement, and also in the correspondence with your editor. Please note that for some data types, deposition in a public repository is mandatory - more information on our data deposition policies and available repositories can be found below: <https://www.nature.com/nature-research/editorial-policies/reporting-standards#availability-of-data>

Nature Structural & Molecular Biology is committed to improving transparency in authorship. As part of our efforts in this direction, we are now requesting that all authors identified as 'corresponding author' on published papers create and link their Open Researcher and Contributor Identifier (ORCID) with their account on the Manuscript Tracking System (MTS), prior to acceptance. This applies to primary research papers only. ORCID helps the scientific community achieve unambiguous attribution of all scholarly contributions. You can create and link your ORCID from the home page of the MTS by clicking on 'Modify my Springer Nature account'. For more information please visit please visit www.springernature.com/orcid.

[Redacted]

Sincerely,
Sara

Sara Osman, Ph.D.
Associate Editor
Nature Structural & Molecular Biology

Reviewers' Comments:

Reviewer #1:

Remarks to the Author:

The manuscript "Structural basis of tRNA recognition by the m³C-RNA methyltransferase METTL6 in complex with SerRS seryl-tRNA synthetase" by Throll, et al. presents a model for how METTL6 and SerRS complex and recognize the tRNA^{Ser} substrate for methylation. High-resolution cryo-EM data help to provide a detailed model the assembly. However, to be able to claim the new findings and interpretations, the authors should address several issues:

1. The structure is of an artificial fusion protein. There are multiple forms of intermolecular contact observed in the structure, with different stoichiometries. How do the authors know that the structure they decided to focus on is the most relevant form? Are the METTL6-SerRS interactions similar in different structures (eg. 1:2:2 vs 2:2:2)? The authors should analyze all the observed interactions and present the compare and contrast, especially if they are depositing the structures. (Since they are depositing all three structures, they should also provide PDV validation reports for all of them.)
2. What is the stoichiometry of the METTL6-SerRS-tRNA complex in solution? Can this information guide the authors how to interpret the different structures? The argument presented with the sentence "The distance suggests that the minimal unit..." is not clear. For example, do the authors think that the 1:1:1 complex (via dimerization mutations on SerRS) and 2:2:2 complex would have the same catalytic activity as 1:2:2? Can they provide any empirical evidence?
3. Can the authors compare the catalytic activity and the affinity for tRNA between the artificial fusion protein vs when METTL6 and SerRS are added separately?
4. At 2.4 angstrom resolution, the authors should provide a more detailed structural analysis showing more atomic interactions. There is only one figure showing the detailed structures with distances far away from the active site. Authors could provide better figures to describe the catalytic pocket and the specific intermolecular interactions. Please clarify how certain residues are chosen to be shown in each figure. Furthermore, motifs such as WDLFYK and FFKDRHW are discussed but not shown in figures.
5. In Figure 2e, it would be helpful to show how the tRNA is moving relative to the proteins. How does the protein complex support the tRNA structural changes?
6. On Page 13, it is not clear why the mutations are chosen. Some of them are not shown in any of the figures.
7. The authors suggest that the structure is for the post-catalytic state, bound with products. Can they discuss how the product release might occur?
8. There seems to a preferred orientation problem on their grids. Can the authors discuss how they have addressed this problem?

Reviewer #2:

Remarks to the Author:

3-Methylcytosine (m³C), found at position 32 of specific tRNAs, is important for tRNA translation efficiency and fidelity. In mammals, METTL6 methylates C32 of cytosolic tRNA^{Ser}, and the methylation of tRNA^{Ser} depends on ser-tRNA synthetase, SerRS. In this

manuscript, Throll and Kowalinski et al. present the CryoEM structure of the METTL6-SerRS-tRNASer complex. This structure reveals the detailed mechanism by which METTL6 specifically recognizes tRNASer and transfers a methyl group to C32 of tRNASer, along with SerRS. The cryoEM structures, as well as the crystal structures presented in the manuscript, are solid, and the manuscript is well-organized and presented. This reviewer highly appreciates the CryoEM structure of the complex and is impressed by the mechanism through which METTL6 specifically recognizes its substrate tRNASer, together with SerRS. This study advances our understanding of the mechanism of the modification enzyme, along with other cofactors, to fulfill its substrate specificity. More importantly, the present work inspires further exploration of the molecular evolution of modification enzymes and aminoacyl-tRNA synthetases.

This review strongly recommends the publication of this manuscript in Nature Structural & Molecular Biology and includes several minor comments that might help the authors improve the manuscript."

Minor comments

1. Page 3 line 60: mt-tRNASer should be mt-tRNASer(UCN) or mt-tRNASerUGA
2. Page4 lines 70-71: other studies suggested that METTL6 activity depends on seryl-tRNA synthetase (SerRS). Cite the references here.
3. Page5. Line90: "stimulation of METTL6 activity bySerRS". The assay conditions for methylation of tRNA in the presence of SerRS (SerRS + Ser +ATP) were not clearly described in the manuscript (page 26 lines 459-471). The authors could kindly describe the assay conditions for the methylation of tRNA in the presence of SerRS.
4. Page5, Line 90: Although the authors claimed that SerRS stimulates METTL6 activity, it is not clear whether SerRS impacts on the affinity of METTL6 for tRNASer and/or accelerates the catalysis. If the authors provide some kinetics data, it will make the manuscript better one.
5. Figure 1b: In the presence of SerRS and serine and ATP, the methylation of tRNASer more efficiently stimulated that on in the presence of SerRS. T It can accelerate the turn-over of the METTL6? The author could kindly discuss this in the text.
6. Figure 1 legend: line5, "that are unresolved in the structure are coloured in gray" should be "that are unresolved in the structure are colored in gray (UNE-S)"
7. Page9 line 157: "contributes to the coordination of SAH (Figure 2c)" SAH is not depicted in the Figure 2C. The author could kindly replace the Figure 2C with SAH in it.
8. Page9, line159: "that both bind to the anticodon stem (Figure 2c, d)" should be "that both bind to the anticodon stem (Figure 2c)."
9. Figure 2: Light orange color letters (labels)in the figures are difficult to read. Thus, authors could kindly change them in the darker color. In figure 2a, the labels for $\alpha N1$, $\alpha N2$, αi , $\beta i1$ and $\beta i2$ could be added. In figure 2d, the label for αi could be added.
10. Page11 line 178: The word "Most interestingly" could be removed.
11. Page 12 line 209, 220: This reviewer feels uncomfortable to the wording "coordinate" in the text. The word could be replaced with "hydrogen-bonds" or other more appropriate words.
12. Page12, line 209: "two isopentane-coordinating arginine residues are conserved throughout .." could be " two isopentane-coordinating arginine residues (R149 and R255) are conserved throughout".
13. Page13 line227: "ultra-conserved" could be "well-conserved".
14. Figure 3C: According to the text in Page 13, Y190F and Y49F in Figure 3c should be Y190A and Y49A, respectively.
15. Page 16 line 262: "recognition factor for METTL and the" should be "recognition factor

for METTL6 and the "

16. This reviewer suggests the inclusion of evolutionary perspectives on the specificity of tRNA modification enzyme with the aid of aminoacyl-tRNA synthetase could make the study more general.

17. Page 24 line425: "PUC19 vector between T7 promoter and BstNI cleavage site "This sentence could be kindly rephrased. It is misleading. pUC19 vector does not have T7 promoter nor BstNI site.

18. Page24 lines 427-428: "40mM TRIS" should be "40 mM Tris-Cl". "0.01% Triton X100" should be 0.01% (w/w) Triton X100".

19. Page26 line463: 5ug should be 5 microg.

20. As pointed out in 3, the assay conditions are not well described. The authors should kindly provide the details of the assay conditions (volume of the assay and so on) to enable the other researchers to reproduce the experiments.

Author Rebuttal to Initial comments

Dear Reviewer,

Thank you for your valuable time and thought reviewing our manuscript. In the response below, we address the remaining concerns.

Best wishes,
Eva Kowalinski

Reviewers' Comments:

Reviewer #1:

I support the publication of this revised manuscript, but my recommendation still stands. I think the authors misunderstood the previous comments. This is a simple recommendation that is typical for structure figures. The choice of visualized side chains should be clearly described in the figure legend so that the reader can objectively assess the structure, without the author cherry-picking the contacts they want to highlight. This can be as simple as defining the radial distance in the figure legend. The revision table 1 is not helpful.'

We have added the following sentences to the figure legend:

Figure 2: "Residues involved in C₃₂ binding and residues of the FFKDRHW motif are represented with side chains as sticks." and "Residues involved in U₃₄ or i⁶A₃₇ binding are represented with side chains as sticks."

Figure 3: "Residues involved in i⁶A₃₇ coordination are represented with side chains as sticks."

Figure 4: "Residues within a 5 Å radius of SAH are represented with side chains as sticks." and "The m³C₃₂ binding site next to the SAH binding site. The m³C₃₂ binding site adjacent to the SAH binding site. Residues within a 5 Å radius of m³C₃₂ are represented with side chains as sticks."

Figure 5: "Residues involved in hydrogen with U_{47D} are represented with side chains as sticks." and "Conserved residues within the METTL6-SerRS interface are represented with side chains as sticks."

Final Decision Letter:

Decision Letter, first revision:

Message: 13th Nov 2023

Dear Dr. Kowalinski,

Thank you again for submitting your manuscript "Structural basis of tRNA recognition by the m³C-RNA-methyltransferase METTL6 in complex with SerRS seryl-tRNA synthetase". We now have comments (below) from the 2 reviewers who evaluated your paper. In light of those reports, we remain interested in your study and would like to see your response to the comments of the referees, in the form of a revised manuscript.

You will see that while both reviewers appreciate the manuscript is improved, reviewer #1 still has outstanding concerns about the stoichiometric composition of the METTL6 SerRS complex, and proposes measurements to explain the proposed cooperativity model. Please be sure to address/respond to all concerns of the referees in full in a point-by-point response and highlight all changes in the revised manuscript text file. If you have comments that are intended for editors only, please include those in a separate cover letter.

We expect to see your revised manuscript within 6 weeks. If you cannot send it within this time, please contact us to discuss an extension; we would still consider your revision, provided that no similar work has been accepted for publication at NSMB or published elsewhere.

Reporting Summary:

When submitting the revised version of your manuscript, please pay close attention to our [href="https://www.nature.com/nature-portfolio/editorial-policies/image-integrity">Digital Image Integrity Guidelines](https://www.nature.com/nature-portfolio/editorial-policies/image-integrity). and to the following points below:

-- that unprocessed scans are clearly labelled and match the gels and western blots presented in figures.

-- that control panels for gels and western blots are appropriately described as loading on sample processing controls
-- all images in the paper are checked for duplication of panels and for splicing of gel lanes.

Please note that all key data shown in the main figures as cropped gels or blots should be presented in uncropped form, with molecular weight markers. These data can be aggregated into a single supplementary figure item. While these data can be displayed in a relatively informal style, they must refer back to the relevant figures. These data should be submitted with the final revision, as source data, prior to acceptance, but you may want to start putting it together at this point.

Data availability: this journal strongly supports public availability of data. All data used in accepted papers should be available via a public data repository, or alternatively, as Supplementary Information. If data can only be shared on request, please explain why in your Data Availability Statement, and also in the correspondence with your editor. Please note that for some data types, deposition in a public repository is mandatory - more information on our data deposition policies and available repositories can be found below: <https://www.nature.com/nature-research/editorial-policies/reporting-standards#availability-of-data>

While we encourage the use of color in preparing figures, please note that this will incur a

charge to partially defray the cost of printing. Information about color charges can be found at <http://www.nature.com/nsmb/authors/submit/index.html#costs>

Nature Structural & Molecular Biology is committed to improving transparency in authorship. As part of our efforts in this direction, we are now requesting that all authors identified as 'corresponding author' on published papers create and link their Open Researcher and Contributor Identifier (ORCID) with their account on the Manuscript Tracking System (MTS), prior to acceptance. This applies to primary research papers only. ORCID helps the scientific community achieve unambiguous attribution of all scholarly contributions. You can create and link your ORCID from the home page of the MTS by clicking on 'Modify my Springer Nature account'. For more information please visit please visit www.springernature.com/orcid.

[Redacted]

Sincerely,
Sara

Sara Osman, Ph.D.
Associate Editor
Nature Structural & Molecular Biology

Reviewers' Comments:

Reviewer #1:

Remarks to the Author:

The authors have made improvements to the manuscript. I still recommend that the authors address these issues:

1. For all figures with side chains on ribbons, they should specify how the residues were chosen to show the side chain. This includes main figures 2, 3, and 4.
2. The molar ratio of SerRS to METTL6 in cells does not fully answer the question regarding the composition of the complex as unattached proteins. The authors could offer a model with individual affinities. If the authors are proposing cooperativity between SerRS and METTL6, it would help to have measurements. It is reasonable for the authors to argue that they are focusing on the conserved contacts observed in different structures. However, it would improve the manuscript to point out what makes each interaction or conformation different to gain more insight into the larger complexes. Is there symmetry among the protomers? Are the two binding sites for METTL6 equivalent? Is there a

reasonable explanation for why the 1:2:2 complex was resolved to significantly higher resolution? This is unclear from the figures and the text as provided.

*Data table for cryo-EM and crystallography is missing.

Reviewer #2:

Remarks to the Author:

The authors have addressed my concerns and the paper is ready for publication.

Author Rebuttal, first revision:

Dear Reviewers,

Thank you for your valuable time and thought reviewing our manuscript on behalf of all authors. In the responses below, we addressed the remaining concerns.

Best wishes,
Eva Kowalinski

Reviewers' Comments:

Reviewer #1:

Remarks to the Author:

The authors have made improvements to the manuscript. I still recommend that the authors address these issues:

1. For all figures with side chains on ribbons, they should specify how the residues were chosen to show the side chain. This includes main figures 2, 3, and 4.

We revised the display of side chains and their mentions in the text for figures 2-5. The rationale for our choices is listed in revision table 1, appended to this file. Corrections in the text are marked with track changes.

2. The molar ratio of SerRS to METTL6 in cells does not fully answer the question regarding the composition of the complex as unattached proteins.

In theory, six different tRNA-containing complexes can exist in equilibrium in solution (Revision Figure 1). Due to the equilibrium between these states and the transient (weak affinity) nature of the interaction with METTL6, we could not determine a distinct in-solution stoichiometry of the complex [*]. We discuss our view on the possible physiological stoichiometry in the discussion section. Future experiments will address the question in the physiological context of the cell.

Revision Figure 1: All possible tRNA-containing complexes that can exist in equilibrium in solution. METTL6 is coloured orange, one protomer of SerRS is coloured purple and the second is coloured salmon, tRNA is coloured grey.

[*] Our attempts to measure the in-solution stoichiometry:

1. Mass photometry requires a sample concentration of ~50 nM concentration, which is too diluted to observe a complex.
2. Size exclusion coupled with multiple-angle laser light scattering (SEC-MALLS) is not applicable. We can only form the complex on a Superdex 200 Increase 3.2/300 column (column volume of 2.4 ml). On the larger column, which is required for SEC-MALLS (Superdex 200 Increase 30/300 column, 24 ml column volume), the sample is too diluted to observe a complex.
3. UC: inconclusive, suggests an association dissociation equilibrium between the components.
4. Native gel: inconclusive mixture

The authors could offer a model with individual affinities.

We determined a very weak affinity (~10 μ M) of METTL6 to tRNA (Extended Data Figs 10c and 11b). The affinity of SerRS to tRNA has been previously estimated to 370 nM (Xu, Shi, and Yang 2013). Determining affinities within the holo-complex is technically challenging due to the stoichiometric heterogeneity.

To address the reviewer's suggestion, we attempted to estimate the affinity between METTL6 and the SerRS-tRNA complex by EMSA and fluorescence polarisation. However, the data obtained was noisy and inconclusive.

The observation that all tested point mutants (tRNA binding, SAM binding, and SerRS interface) disrupt the complex and/or reduce methylation activity points towards a weak affinity of METTL6 to the rest of the complex. We discuss this in paragraph three of the discussion.

Reference: Xu, Xiaoling, Yi Shi, and Xiang-Lei Yang. 2013. "Crystal Structure of Human Seryl-tRNA Synthetase and Ser-SA Complex Reveals a Molecular Lever Specific to Higher Eukaryotes." *Structure* 21 (11): 2078–86.

If the authors are proposing cooperativity between SerRS and METTL6, it would help to have measurements.

Our manuscript provides the first full evidence for the METTL6-SerRS-tRNA complex, thus providing the basis for further future detailed kinetic analyses. The term "cooperative" is not mentioned in the manuscript to avoid misinterpretation.

It is reasonable for the authors to argue that they are focusing on the conserved contacts observed in different structures. However, it would improve the manuscript to point out what makes each interaction or conformation different to gain more insight into the larger complexes. Is there symmetry among the protomers? Are the two binding sites for METTL6 equivalent?

Page 7: Processing of the cryo-EM micrographs and particle images of the METTL6-SerRS-tRNA^{Ser} complex revealed 3D classes for 1:2:1, 1:2:2 and 2:2:2 stoichiometries with essentially identical intermolecular contacts in the METTL6-SerRS-tRNA interface (Extended Data fig. 2b, 3a-c).

We compare the data in the form of the coulomb densities of the molecular interfaces. These are identical within the given resolution range (Extended Figure 3). Comparing the atomic models would not be appropriate; these would be identical since they are derived from the highest-resolution data. In the 2:2:2 complex, both METTL6 interfaces are identical given the C2 symmetry (indicated in the processing scheme, Extended Figure 2).

Is there a reasonable explanation for why the 1:2:2 complex was resolved to significantly higher resolution? This is unclear from the figures and the text as provided.

Different factors, like particle orientation, ice thickness, flexibility of the assembly, and number of particles, technically limit the resolution of cryo-EM structures. However, the resolution or number of particles on a cryo-EM grid does not necessarily reflect biological relevance.

*Data table for cryo-EM and crystallography is missing.

We added the missing pages to the Extended Data.

Reviewer #2:

Remarks to the Author:

The authors have addressed my concerns and the paper is ready for publication.

Revision Table 1: Reasoning for the side chains shown in figures

Figure	Residue	Reasoning	Text mention
2d	F56, F57, K58, D59, R60, H61, and W62	Residues forming the conserved FFKDRHW motif on the substrate binding loop	Page 9
	F158, R213	Involved in C32 binding	Page 14
	Q214	Q214 well resolved in map, pointing towards RNA. It would be odd to show the density without it.	Page 18, typo corrected
2e	H195	Involved in A37 binding, shown in detail in Figure 3b	Page 12
	R199	Involved in A37 binding, shown in detail in Figure 3b	Page 12
	R246	The conserved residue R246 is part of the β -extension, interacts with U34 and is well resolved in the density.	Page 9, "This hairpin region carries mostly positively charged residues". The resolution of most other residues in the hairpin does not allow the display of side-chain positions.
	T248	Involved in A37 binding, shown in more detail in Figure 3b	Page 12
	R259	Involved in A37 binding, shown in more detail in Figure 3b	Page 12
3b	H195	Involved in A37 binding, detailed version of Figure 2e	Page 12
	R199	Involved in A37 binding, detailed version of Figure 2e	Page 12
	V247	Involved in A37 binding, detailed version of Figure 2e	Page 12
	T248	Involved in A37 binding, detailed version of Figure 2e	Page 12

	R259	Involved in A37 binding, detailed version of Figure 2e	Page 12
4b	W45	Involved in the coordination of SAH	Page 13
	Y49	Involved in the coordination of SAH, residue mutated	Page 14
	R60	Residues forming the conserved FFKDRHW motif on the substrate binding loop	Page 9
	GCG motif	Involved in the coordination of SAH	Page 13
	D110	Involved in the coordination of SAH, point mutant tested	Page 13
	F111	Involved in the coordination of SAH	Page 13
	L137	Involved in the coordination of SAH	Page 13
	T138	Involved in the coordination of SAH	Page 13
	I157	Involved in the coordination of SAH	Page 13
	F158	Involved in C32 binding	Page 14
	V159	Involved in the coordination of SAH	Page 13
	A162	Involved in the coordination of SAH	Page 13
	V163	Involved in the coordination of SAH	Page 13
4c	W45	Involved in the coordination of SAH and C32 binding	Pages 13 and 14
	F57	Involved in C32 binding	Page 14
	R60	Involved in C32 binding	Page 14
	W62	Involved in C32 binding	Page 14

	F158	Involved in C32 binding	Page 14
	Y190	Involved in C32 binding, point mutant tested	Page 14
	R213	Involved in C32 binding	Page 14
	R259	Involved in C32 binding	Page 14
	F261	Involved in C32 binding	Page 14
5	F32	Involved in recognition of SerRS, point mutant tested	Page 17
	K43	Involved in variable arm recognition	Page 16: mention added
	D46	Involved in variable arm recognition	Page 17
	K50	Involved in variable arm recognition	Page 17
	K51	Involved in recognition of SerRS	Page 17
	R114	Involved in variable arm recognition	Page 17
	D215	Involved in recognition of SerRS	Page 17, typo corrected
	SerRS K12	Involved in backbone interaction with METTL6	Page 17
	SerRS G13	Involved in backbone interaction with METTL6	Page 17
	SerRS M416	Involved in interaction with METTL6, point mutant tested	Page 17
	SerRS M417	Involved in interaction with METTL6, point mutant tested	Page 17

Decision Letter, second revision:

Message: Our ref: NSMB-A47884B

18th Jan 2024

Dear Dr. Kowalinski,

Thank you for submitting your revised manuscript "Structural basis of tRNA recognition by the m³C-RNA-methyltransferase METTL6 in complex with SerRS seryl-tRNA synthetase" (NSMB-A47884B). It has now been seen by the original referees and their comments are below. The reviewers find that the paper has improved in revision, and therefore we'll be happy in principle to publish it in Nature Structural & Molecular Biology, pending minor revisions to satisfy the referees' final requests and to comply with our editorial and formatting guidelines.

We are now performing detailed checks on your paper and will send you a checklist detailing our editorial and formatting requirements in the next few weeks. Please do not upload the final materials and make any revisions until you receive this additional information from us.

To facilitate our work at this stage, it is important that we have a copy of the main text as a word file. If you could please send along a word version of this file as soon as possible, we would greatly appreciate it; please make sure to copy the NSMB account (cc'ed above).

Sincerely,
Sara

Sara Osman, Ph.D.
Associate Editor
Nature Structural & Molecular Biology

Reviewer #1 (Remarks to the Author):

I support the publication of this revised manuscript, but my recommendation still stands. I think the authors misunderstood the previous comments. This is a simple recommendation that is typical for structure figures. The choice of visualized side chains should be clearly described in the figure legend so that the reader can objectively assess the structure, without the author cherry-picking the contacts they want to highlight. This can be as simple as defining the radial distance in the figure legend. The revision table 1 is not helpful.

Author Rebuttal, second revision:

Message: 29th May 2024

Dear Dr. Kowalinski,

We are now happy to accept your revised paper "Structural basis of tRNA recognition by the m³C-RNA-methyltransferase METTL6 in complex with SerRS seryl-tRNA synthetase" for publication as an Article in Nature Structural & Molecular Biology.

Your paper will be published online soon after we receive proof corrections and will appear in print in the next available issue. You can find out your date of online publication by contacting the production team shortly after sending your proof corrections.

Please note that *Nature Structural & Molecular Biology* is a Transformative Journal (TJ). Authors may publish their research with us through the traditional subscription access route or make their paper immediately open access through payment of an article-processing charge (APC). Authors will not be required to make a final decision about access to their article until it has been accepted. Find out more about Transformative Journals

Sincerely,
Sara

Sara Osman, Ph.D.
Senior Editor
Nature Structural & Molecular Biology

Dear Reviewers,

In the name of all authors, I would like to thank you for your valuable time to review our manuscript. In the responses below, we addressed all your concerns and believe that your contributions improved the scientific quality of our manuscript and especially the comprehensiveness of our figures.

Best wishes,
Eva Kowalinski

Reviewers' Comments:

Reviewer #1:

Remarks to the Author:

The manuscript "Structural basis of tRNA recognition by the m³C-RNA methyltransferase METTL6 in complex with SerRS seryl-tRNA synthetase" by Throll, et al. presents a model for how METTL6 and SerRS complex and recognize the tRNA^{Ser} substrate for methylation. High-resolution cryo-EM data help to provide a detailed model the assembly. However, to be able to claim the new findings and interpretations, the authors should address several issues:

In the comments, reviewer #1 has raised several concerns regarding the use of an artificial fusion protein. I would like to clarify that all functional data in our study have been generated using individually purified proteins. The fusion construct was employed solely for structure determination. Here are several points that underscore the physiological relevance and accuracy of our model:

- Linker flexibility: The linker used in the fusion protein is long and flexible. In fact, no coulomb density is present for the 61 amino acids of the linker. This flexibility ensures that the linker does not impose structural constraints on the protein-protein interactions.
- No enforced protein-protein contact: It is important to note that the linker does not enforce the protein-protein contact. Throughout all 3D classes of the cryo-EM sample, we have observed free SerRS molecules without associated METTL6 density, indicating that METTL6 can freely dissociate from the complex.
- Multiple observations of the same contacts: In our cryo-EM samples, we observe four distinct trimeric METTL6-SerRS-tRNA entities. These include two trimers in the 2:2:2 complex, one trimer in the 1:2:2 complex, and one trimer in the 1:1:1 complex. Importantly, our coulomb density maps reveal similar contact points among these entities. We have added the maps of the contact regions in Extended Data Fig. 3).
- Conserved protein-protein interfaces: Our model is further supported by multi-species sequence alignments (Extended data 5 and Figures 11g,h), which demonstrate that the protein-protein and protein-RNA interfaces are conserved across different species.
- Functional impact of mutations: We have also conducted experiments with individually purified mutant proteins. These mutations disrupt the complex (see Extended Figure 11e).

1. The structure is of an artificial fusion protein. There are multiple forms of intermolecular contact observed in the structure, with different stoichiometries. How do the authors know that the structure they decided to focus on is the most relevant form? Are the METTL6-SerRS interactions similar in different structures (eg. 1:2:2 vs 2:2:2)? The authors should analyze all the observed interactions and present the compare and contrast, especially if they are depositing the structures. (Since they are depositing all three structures, they should also provide PDV validation reports for all of them.)

Our cryo-EM dataset yielded three different stoichiometries of the METTL6-SerRS-tRNA complex, containing all together four METTL6-SerRS-tRNA trimeric entities. The contacts within each of the trimers are identical as judged by the cryo-EM maps. We added Extended Data figure 3 showing a closeup of the maps in the contact region.

With this revision, we also provide validation reports for all structures.

2. What is the stoichiometry of the METTL6-SerRS-tRNA complex in solution? Can this information guide the authors on how to interpret the different structures? The argument presented with the sentence "The distance suggests that the minimal unit..." is not clear. For example, do the authors think that the 1:1:1 complex (via dimerization mutations on SerRS) and 2:2:2 complex would have the same catalytic activity as 1:2:2? Can they provide any empirical evidence?

The complex is very fragile and we have technical difficulties measuring its stoichiometry in solution (*). However, we do not believe the in-solution stoichiometry would reflect the physiological reality in the cell: according to a quantitative mass spectrometry study, aaRS are 100 times more abundant than METTL6 (Hein et al., 2015), making a complex that contains METTL6 in sub-stoichiometry more likely.

SerRS dimer interface mutants will not reveal the desired insights for the following reasons: SerRS forms an obligatory dimer with two active protomers. Each of the one or two bound tRNA molecules makes contact with both SerRS subunits, with the variable arm contacting NTD of the proximal SerRS protomer 1 and the acceptor arm contacting the AD of the distal SerRS protomer. Disruption of the SerRS dimer interface would hence destabilize the tRNA-SerRS interaction. This would inevitably impact the fragile METTL6 interaction.

To be clear about the speculative nature of our claims on stoichiometry, we have removed reflections on the stoichiometry of the METTL6-SerRS-tRNA complex from the results section and rather added a paragraph to the discussion section:

Shortened results section:

"Processing of the cryo-EM micrographs and particle images of the METTL6-SerRS-tRNA^{Ser} complex revealed 3D classes for 1:2:1, 1:2:2 and 2:2:2 stoichiometries with essentially identical intermolecular contacts in the METTL6-SerRS-tRNA interface (Extended Data fig. 2 and. 3). This heterogeneity is partially due to the fact that SerRS forms an obligatory dimer that can bind either one or two tRNAs ³³⁻³⁷. Surprisingly, despite the fusion of METTL6 to SerRS, not all of the SerRS molecules in our reconstructions displayed an associated density for METTL6. In the absence of protease treatment or cleavage sites, this suggests

that the 60 amino acids long flexible linker connecting the proteins allowed free dissociation of METTL6 and that the fusion strategy tolerated an on-off equilibrium of the assembly. For model building and interpretation, we focused on the highest resolution class at 2.4 Å, resolving a 1:2:2 METTL6: SerRS: tRNA complex (Extended Data Table 2)."

Additional paragraph in the discussion section:

"In our samples, we observe different possible stoichiometries of the METTL6-SerRS-tRNA complex. In the complex with two copies of METTL6 present, these are spatially distant (~100 Å) from each other with each METTL6 molecule binding only one tRNA, implying that that these METTL6 molecules would function independently of each other. According to quantitative mass spectrometry, the cellular abundance of SerRS is approximately 100 times higher than METTL6⁶¹. Therefore we would reckon that the most abundant physiological active assembly comprises the SerRS dimer, one or two tRNAs and one copy of METTL6, but does not exclude the association of a second METTL6 molecule to the complex."

(*) To determine the molecular masses of complexes in solution, we would typically use mass photometry which needs ~50 nM concentration (too low to observe a complex) or size exclusion coupled with multiple-angle laser light scattering (SEC-MALLS). While we can observe the complex on a Superdex 200 Increase 3.2/300 column (with a column volume of 2.4 ml), however, the sample is too diluted to observe a complex on a Superdex 200 Increase 30/300 column (with a 24 ml column volume), which is incompatible with common SEC-MALLS equipment due the requirement of a minimal flow rate.

Reference:

Hein, M. Y. et al. A human interactome in three quantitative dimensions organized by stoichiometries and abundances. *Cell* 163, 712–723 (2015).

3. Can the authors compare the catalytic activity and the affinity for tRNA between the artificial fusion protein vs when METTL6 and SerRS are added separately?

The possible outcomes of such an experiment would be that the fusion protein has either more, less or the same activity. We believe that these outcomes would only have limited influence on the interpretation of our data.

The fusion polypeptide served solely for the purpose of stabilizing the assembly on cryo-EM grids. All functional and mutant data of the study are generated with individual purified proteins. The cellular co-purified RNA is fully m³C modified (Extended Data Figure 6) and SAH is present in the structure despite the addition of sinefungine to the sample, hinting towards that the fusion construct is active. However, the stable co-purification of the two products with the fusion construct suggests that this sample has defects in product release.

To make a better distinction between the cryo-EM construct and the individual constructs used for the assays we rephrased in the manuscript:

Initial: "Therefore, we stabilised the assembly by fusing METTL6 to the flexible C-terminus of SerRS and co-expressed both proteins as a single polypeptide in insect cells."

Rephrased: "Therefore, for the purpose of structure determination, we stabilised the assembly by fusing METTL6 to the flexible C-terminus of SerRS and co-expressed both proteins as a single polypeptide in insect cells."

4. At 2.4 angstrom resolution, the authors should provide a more detailed structural analysis showing more atomic interactions. There is only one figure showing the detailed structures with distances far away from the active site. Authors could provide better figures to describe the catalytic pocket and the specific intermolecular interactions. Please clarify how certain residues are chosen to be shown in each figure. Furthermore, motifs such as WDLFYK and FFKDRHW are discussed but not shown in the figures.

RNA binding of the m³C-RBD:

We adapted Figure 2. We added a surface-charge representation of the tRNA-binding pocket in Figure 2c. Since base-specific contacts in the binding groove are absent, we reduced in Figures 2d and 2e the misleading stick representations of the bases to cartoons. In Figures 2d and 2e we also include densities around the protein moieties to visualize how the anticodon arm is nested within the motifs comprising the m³C-RBD.

We added a sentence to the manuscript: "The [m³C-RBD] domain forms a highly positively charged groove that accommodates the anticodon stem through base-unspecific electrostatic contacts with the phosphodiester backbone; only the modification base C₃₂ is recognized through base-specific contacts (Fig. 2c)."

Catalytic pocket:

We adapted Figure 4 (former Figure 3) to depict SAM binding in two views (Figure 4b) and C₃₂ binding (Figure 4c) in separate panels. Since the joint coordination of both substrates is difficult to visualize in 2D, we added the schematic Extended Data figure 9 that puts both binding sites in context.

We added a more detailed description of the SAM binding:

"The cryo-EM map density suggested the presence of SAH in our reconstruction structure. Like in RNA-free crystal structures of METTL6^{31,32}, the methyl donor is embedded in a deep pocket adjacent to the RNA binding cleft in METTL6 (Fig. 4a). Motif I of methyltransferases (EaGCGvGN in METTL6) and the main chain carbonyl of isoleucine I157 interact with the amino acid portion of SAH; arginine R60 forms a salt bridge with the carboxyl group of the co-factor. Aspartate D110 in the acidic loop of motif II, which is highly conserved in SAM-dependent methyltransferases, maintains hydrogen bonds to the sugar hydroxyl groups of SAH, and also tryptophane W45 and valine V159 contribute to the sugar coordination. Phenylalanine F111 of the acidic loop and other mostly hydrophobic residues (A162, V163, F111, L124, T138) form the pocket for the adenosyl moiety of SAM (Fig. 4b, Extended Data fig. 9)."

We added a more detailed description of the C₃₂ coordination:

"The m³C₃₂ nucleotide is tightly coordinated in the interface of the methyltransferase core and the m³C-RBD (Fig. 4c, Extended Data fig. 9). The bottom of the C₃₂ binding pocket is formed by aromatic residues of the methyltransferase core, i.e. tyrosine Y190,

and phenylalanines F261 and F158. Interestingly, F158 lies in the position where some other SAM-dependent RNA and DNA methyltransferases would carry the catalytic (D/N)PP(F/W/Y) motif⁴⁰. Phenylalanines F158 (methyltransferase core) and F57 (m³C-RBD) stack the C₃₂ base via π -interactions. Two opposed arginine residues interact from each side with functional groups of the base R60 (m³C-RBD) via hydrogen bonding with the carbonyl and R213 (core) with the amino group, suggesting a deprotonated state. Several other residues lie in distances that would allow van-der-Waals interactions with C₃₂, such as tryptophan W45 and tyrosine Y190. Adjacent to C₃₂, the phospho-diester backbone of the RNA is stabilized through interactions with tyrosine Y190, arginine R259, phenylalanine F261, and tryptophane W62."

We added indications for WDLFYK and FFKDRHW motifs in all figures featuring these motifs.

We improved consistency and show all residues and motifs mentioned in the text in the figures and vice-versa.

i⁶A37 coordination:

We have included a new Figure 3b that better describes the coordination of the A37 modification.

METTTL6-SerRS contact loops:

Two sentences modified:

Initial: But remarkably only two loops of SerRS interact directly with METTTL6 (Fig. 5c, d). In the first contact point, the main chain of an NTD loop of SerRS contacts the side chain of phenylalanine F32 in the m³C-RBD (Fig. 5c and Extended Data fig. 3a and 11c).

Rephrased: "But remarkably only two loops of SerRS interact directly with METTTL6 (Fig. 5c, d). In the first contact point, the main chain of an NTD loop of SerRS (lysine K12 and glycine G13) contacts the side chain of phenylalanine F32 in the m³C-RBD (Fig. 5c and Extended Data fig. 3a and 11c)."

Added sentence: "Here, polar interactions of two SerRS-AD methionines, M416 and M417, engage with METTTL6 m³C-RBD elements, particularly via arginine R51 and main-chain interactions to a β -hairpin in the β -insertion (residues N214-G116)."

5. In Figure 2e, it would be helpful to show how the tRNA is moving relative to the proteins. How does the protein complex support the tRNA structural changes?

We have added the Extended Data figure 6c that places the RNA distortion in the context of its orientation relative to METTTL6.

6. On Page 13, it is not clear why the mutations are chosen. Some of them are not shown in any of the figures.

We have added a paragraph to the manuscript describing our reasoning for the mutations. The reasoning might also be more evident with the 2D schematic of the active site that we have added in Extended Data Figure 9 (see also comment 4).

Added paragraph:

"To probe the METTL6 active site experimentally, we tested single-point mutants for enzymatic activity, methyl-donor and tRNA^{Ser}_{UGA} binding, and complex formation (Fig. 4d, Extended Data fig. 10a-d). SAM-dependent methyltransferases use diverse reaction mechanisms⁵². The absence of cysteines in proximity to C₃₂, excludes the possibility of covalent enzyme-nucleotide intermediate formation with the base as it had been described for the uracil-5-methyltransferase TrmA⁵³. Glutamates or aspartates that could be potentially catalytically relevant are not found in the close environment of C₃₂. Therefore, we tested mutations of two conserved tyrosines in proximity to C₃₂ as potential candidates that could play a role in enzyme activity."

7. The authors suggest that the structure is for the post-catalytic state, bound with products. Can they discuss how the product release might occur?

Our study does not provide experimental evidence for how the products are released. We envisage future research, including molecular dynamics simulations, to reveal a detailed reaction mechanism. For METTL8, Kleiber et al., have measured the affinity of METTL8 to RNA hairpins comprising m³C or not and speculate that the reduced affinity of the product might help to release the products. However, Kleiber et al., do not use a full tRNA or mt-SerRS co-factor, nor include the second reaction product SAH, therefore these results might not be directly transferable to METTL6.

8. There seems to be a preferred orientation problem on their grids. Can the authors discuss how they have addressed this problem?

The reviewer is correct that in the 2:2:1 dataset, one orientation is more represented than others. However, this did not impose problems in data processing and enabled a high-quality reconstruction to 2.4 Å resolution without evident anisotropy of the final map. Therefore we did not specifically address the problem. Good map quality is evident from the Extended Data figures 3, 4b-c, 7c, 11c-d.

Reviewer #2:

Remarks to the Author:

3-Methylcytosine (m³C), found at position 32 of specific tRNAs, is important for tRNA translation efficiency and fidelity. In mammals, METTL6 methylates C32 of cytosolic tRNA^{Ser}, and the methylation of tRNA^{Ser} depends on ser-tRNA synthetase, SerRS. In this manuscript, Throll and Kowalinski et al. present the CryoEM structure of the METTL6-SerRS-tRNA^{Ser} complex. This structure reveals the detailed mechanism by which METTL6 specifically recognizes tRNA^{Ser} and transfers a methyl group to C32 of tRNA^{Ser}, along with SerRS. The cryoEM structures, as well as the crystal structures presented in the manuscript, are solid, and the manuscript is well-organized and presented. This reviewer highly appreciates the CryoEM structure of the complex and is impressed by the mechanism through which METTL6 specifically recognizes its substrate tRNA^{Ser}, together with SerRS.

This study advances our understanding of the mechanism of the modification enzyme, along with other cofactors, to fulfill its substrate specificity. More importantly, the present work inspires further exploration of the molecular evolution of modification enzymes and aminoacyl-tRNA synthetases.

This review strongly recommends the publication of this manuscript in Nature Structural & Molecular Biology and includes several minor comments that might help the authors improve the manuscript."

Minor comments

1. Page 3 line 60: mt-tRNA^{Ser} should be mt-tRNA^{Ser}(UCN) or mt-tRNA^{Ser}UGA

Corrected.

2. Page 4 lines 70-71: other studies suggested that METTL6 activity depends on seryl-tRNA synthetase (SerRS). Cite the references here.

References included.

3. Page 5. Line 90: "stimulation of METTL6 activity by SerRS". The assay conditions for methylation of tRNA in the presence of SerRS (SerRS + Ser + ATP) were not clearly described in the manuscript (page 26 lines 459-471). The authors could kindly describe the assay conditions for the methylation of tRNA in the presence of SerRS.

A comprehensive description of the assay has been included in the materials and methods section.

Rephrased:

"Methyltransferase assays were performed in 6 mM HEPES-KOH (pH 7.9), 0.4 mM EDTA, 10 mM DTT, 80 mM KCl, 1.5 mM MgCl₂, RNasin (40U/μl) (Promega), and 1.6% glycerol in a total volume of 200 μl. Proteins (METTL6, SerRS) were used in 0.05 μM final concentration. As substrate, 0.4 μM of in vitro transcribed tRNA^(ser) or tRNA^(thr), or 5 μg of total RNA extracted from HeLa cells was added. 4 μM of Serine (Sigma) and 2 mM of ATP were added where indicated. 1 μl of 3H-labelled SAM (1 mCi/ml; PerkinElmer) was added to the mixture and then incubated at 30°C for 1 h with gentle shaking. Then another 1 μl of 3H-SAM was added and the assay continued at room temperature (22°C) overnight followed by column purification of the RNA with the Zymo Research RNA Miniprep kit (used according to the manufacturer's instructions). After elution of RNA from the columns in 150 μl nuclease-free H₂O, 500 μl of AquaLight liquid scintillation counter cocktail (Hidex) was added and tritium incorporation analysed by scintillation counting with a Triathler Counter (Hidex). All data of in vitro methyltransferase assays are shown as blank-subtracted mean of three scintillation counts, and three experimental replicates that were carried out for each experiment."

4. Page 5, Line 90: Although the authors claimed that SerRS stimulates METTL6 activity, it is not clear whether SerRS impacts on the affinity of METTL6 for tRNA^{Ser} and/or accelerates the catalysis. If the authors provide some kinetics data, it will make the manuscript better one.

The reviewer has requested a comparison of the kinetics of METTL6-tRNA reactivity in the presence and absence of SerRS. We have considered this request, but our laboratories are encountering some technical limitations. Specifically, one of our labs lacks authorization to work with tritium, while the other lab lacks a suitable setup for phenol-chloroform handling, which is necessary for quickly stopping reactions for a time-course study. Unfortunately, the colourimetric assay available (MTase-Glo™, Promega) lacks the sensitivity required to detect METTL6 activity when SerRS is absent.

In the suggested experiment, the substantial 100x difference in activity observed in the endpoint assay between the +/- SerRS conditions would require to employ very different experimental setups for each condition (e.g. different reactant concentrations, different choice of time intervals). Thus, these different experimental conditions would raise concerns about the comparability of the generated catalytic parameters between the two conditions.

As an alternative to approaching the reviewers' comment, we have attempted a fluorescence polarisation assay with fluorescently labelled tRNA to determine the affinity of METTL6 to the tRNA-SerRS complex, which did not result in interpretable curves: while upon binding the polarisation signal augments indicating binding, the curves are linear, making it impossible to deduct binding affinities. The measurements may be impacted to a certain degree by the on-off equilibrium within the SerRS-tRNA complex. Additionally, the different possible stoichiometries in the complex might interfere with proper data generation as well. We thus prefer not to include these data. Future molecular dynamics simulations that we are planning might reveal insights into the details of the catalytic mechanism. Accounting for the fragility of the complex, however, we would hypothesize that the affinity of METTL6 to tRNA is not greatly enhanced through the presence of SerRS.

Based on this reviewer's comment, we add the following section to the discussion:

"We observe that the association of METTL6 to the SerRS-tRNA complex is rather labile, exemplified through the dissociation of METTL6 from the complex in cryo-EM samples, complex dissociation in higher dilutions (e.g. on SEC columns with larger volume) and the interference of single point mutations with complex formation (Fig. 4d, 5e). Therefore, we speculate that the involvement of SerRS does not necessarily provide a higher affinity towards tRNA but rather that the complex provides the necessary geometry to trigger the reaction."

5. Figure 1b: In the presence of SerRS and serine and ATP, the methylation of tRNA^{Ser} more efficiently stimulated than on in the presence of SerRS. T It can accelerate the turn-over of the METTL6? The author could kindly discuss this in the text.

According to quantitative mass spectrometry, the cellular abundance of SerRS is approximately 100 times higher than METTL6 (Hein et al., 2015) Therefore, we speculate that under physiological conditions, METTL6 primarily acts on tRNAs that have previously encountered SerRS and undergone aminoacylation. In our methyltransferase assay, we provoked tRNA serylation through the addition of ATP and serine, which induced enhanced activity. This suggests, even though METTL6 can catalyze reactions with non-serylated in vitro transcribed tRNA, the complex geometry might have evolved to be optimal for serylated tRNAs. It is worth noting that the study by Mao et al., 2021 also indicates that the activity of

SerRS is not necessary for METTL6 activity, as tested with a catalytic inactive mutant of SerRS.

Based on this reviewer's comment, we have added a paragraph in the discussion section:

"It has been demonstrated that METTL6 activity does not depend on the serylation activity of SerRS ²⁰. Although we did not collect evidence on the order of the catalytic events under physiologic conditions, we would anticipate that serylation precedes methylation with the following reasoning: METTL6 is active on in vitro transcribed non-serylated tRNA; based on the cellular protein ratio with a large excess of SerRS in respect to METTL6, the tRNA is likely to encounter SerRS prior to interaction with METTL6; additionally, we observe the coordination of a another RNA modification, i^6A_{37} or t^6A_{37} , suggesting that m^3C -methylation is a late event in the tRNA biogenesis. The increased activity of METTL6, when ATP and serine are added to our assay (Fig. 1b), may be attributed to the favorable geometric alignment of the m^3C -methylation complex, which appears to be ideal for serylated tRNA^{Ser}, the probable physiologic substrate."

Reference:

Hein, M. Y. et al. A human interactome in three quantitative dimensions organized by stoichiometries and abundances. *Cell* 163, 712–723 (2015).

6. Figure 1 legend: line5, "that are unresolved in the structure are coloured in grey "should be "that are unresolved in the structure are coloured in grey (UNE-S)"

Regions unresolved in the structure comprise both N- and C-terminal regions of METTL6 and the UNE-S of SerRS. We added this information to the figure legend.

7. Page9 line 157: "contributes to the coordination of SAH (Figure 2c)" SAH is not depicted in the Figure 2C. The author could kindly replace the Figure 2C with SAH in it.

Including SAH in this view would make the figure less comprehensive (SAH would lie in front of everything). Therefore, we prefer here to additionally to refer to Figure 2a, which depicts SAH, and add a more detailed interaction to SAH in Figure 4b.

Initial: "The substrate binding loop is central in coordinating the target base C_{32} and contributes to the coordination of SAH (Figure 2c)."

Rephrased: "The substrate binding loop is central in coordinating the target base C_{32} and contributes to the coordination of SAH through arginine R60 (Figure 2a, 2c)."

8. Page9, line159: "that both bind to the anticodon stem (Figure 2c, d) "should be "that both bind to the anticodon stem (Figure 2c)."

We had overlooked labelling the turn in Figure 2d as part of the insertion. We have added the label "insertion domain" to Figure 2d.

9. Figure 2: Light orange color letters (labels)in the figures are difficult to read. Thus, authors could kindly change them in the darker color. In figure 2a, the labels for N1,N2, i, i1 and i2 could be added. In figure 2d, the label for i could be added.

The orange colour of labels was changed to a darker tone in all figures. We have added the missing labels in Figures 2a and 2d.

10. Page11 line 178: The word "Most interestingly "could be removed.

Done.

11. Page 12 line 209, 220: This reviewer feels uncomfortable to the wording "coordinate" in the text. The word could be replaced with "hydrogen-bonds" or other more appropriate words.

In the case of i^6A no hydrogen bonds can form between the isopentyl moiety and the protein. Rather, apolar interactions are formed between the aliphatic modification and the plane of the guanidinium group (Armstrong et al., 2016, Mason et al., 2003). But hydrogen bonding would indeed be involved in the case of the t^6A modification, which seems a requirement for the m^3C -tRNA methyltransferases METTL2 and METTL8. Based on the reviewer's comment we have rephrased the paragraph and included a model of t^6A binding featuring hydrogen bonding with METTL2 and METTL8 alphafold models.

We have rephrased the paragraph:

"Although the A_{37} modification was not a requirement for METTL6 activity ³²⁰, the coordination of this modification in our complex structure suggests that i^6A_{37} might assist in stabilizing the remodelled tRNA^{Ser} configuration. The two isopentyl-binding arginine residues (R199 and R259) are conserved throughout the m^3C methyltransferase family, suggesting that the METTL6 paralogues would coordinate i^6A_{37} or t^6A_{37} in a similar fashion, and possibly exhibit hydrogen bonding towards t^6A (Extended Data fig. 8)."

References:

Armstrong, C.T., Mason, P.E., Anderson, J.L.R., and Dempsey, C.E. (2016). Arginine side chain interactions and the role of arginine as a gating charge carrier in voltage sensitive ion channels. *Sci. Rep.* 6, 21759.

Mason, P.E., Neilson, G.W., Dempsey, C.E., Barnes, A.C., and Cruickshank, J.M. (2003). The hydration structure of guanidinium and thiocyanate ions: implications for protein stability in aqueous solution. *Proc. Natl. Acad. Sci. U. S. A.* 100, 4557–4561.

12. Page12, line 209: "two isopentane-coordinating arginine residues are conserved throughout .." could be " two isopentane-coordinating arginine residues (R149 and R255) are conserved throughout".

Inserted as requested: "The two isopentyl-coordinating arginine residues (R199 and R259)..."

13. Page13 line227: "ultra-conserved "could be "well-conserved ".

Replaced by "highly conserved".

14. Figure 3C: According to the text in Page 13, Y190F and Y49F in Figure 3c should be Y190A and Y49A, respectively.

Y49F and Y190F in the figure are correct. The mistake in the text was corrected accordingly.

15. Page 16 line 262: "recognition factor for METTL and the "should be "recognition factor for METTL6 and the "

Corrected.

16. This reviewer suggests the inclusion of evolutionary perspectives on the specificity of tRNA modification enzyme with the aid of aminoacyl-tRNA synthetase could make the study more general.

We have included an additional paragraph in the discussion and one sentence in the introduction:

Introduction: "This m³C-tRNA-methylation complex reveals a new moonlighting function of an aminoacyl-tRNA synthetase specifically targeting its bonafide set of substrates for chemical base modification."

Discussion: "Aminoacetylation-independent functions of aminoacylases have been identified, most commonly as gene-regulatory auto-feedback mechanisms⁵⁴⁻⁵⁷. SerRS plays a non-canonical role in vascular development that is independent of its aminoacylation activity ⁵⁸⁻⁶⁰. Here we reveal a to our knowledge novel moonlighting function of an aminoacylase as a substrate selection factor. SerRS has high specificity for only tRNA^{Ser} and as such facilitates very selectively the chemical base modification of its own set of isoacceptors."

17. Page 24 line425: "PUC19 vector between T7 promoter and BstN1 cleavage site "This sentence could be kindly rephrased. It is misleading. pUC19 vector does not have T7 promoter nor BstNI site.

Corrected.

Rephrased: The DNA sequences coding for a T7 promoter, human tRNA^{Ser}_{UGA} or tRNA^{Thr}_{CGU}, respectively, directly followed by a BstN1 cleavage site were cloned into a pUC19 vector.

18. Page24 lines 427-428: "40mM TRIS" should be "40 mM Tris-Cl". "0.01% Triton X100" should be 0.01% (w/w) Triton X100".

Corrected.

19. Page26 line463: 5ug should be 5 microg.

Corrected.

20. As pointed out in 3, the assay conditions are not well described. The authors should kindly provide the details of the assay conditions (volume of the assay and so on) to enable the other researchers to reproduce the experiments.

Corrected. See comment 3.